# TRAIN (Transcription of Repeats Activates INterferon) in response to chromatin destabilization induced by small molecules in mammalian cells

Katerina Leonova[1], Alfiya Safina[1], Elimelech Nesher[1,2], Poorva Sandlesh[1], Rachel Pratt[1], Catherine Burkhart[3], Brittany Lipchick[1], Ilya Gitlin[1], Costakis Frangou[1], Igor Koman[2], Jianmin Wang[4], Kirill Kirsanov[5], Marianna G Yakubovskaya[5], Andrei V Gudkov[1], Katerina Gurova[1]*

[1]Department of Cell Stress Biology, Roswell Park Cancer Institute, Buffalo, United States; [2]Department of Molecular Biology, Ariel University, Ariel, Israel; [3]Buffalo BioLabs, Buffalo, United States; [4]Department of Bioinformatics, Roswell Park Cancer Institute, Buffalo, United States; [5]Department of Chemical Carcinogenesis, Institute of Carcinogenesis, Blokhin Cancer Research Center RAMS, Moscow, Russia

**Abstract** Cellular responses to the loss of genomic stability are well-established, while how mammalian cells respond to chromatin destabilization is largely unknown. We previously found that DNA demethylation on p53-deficient background leads to transcription of repetitive heterochromatin elements, followed by an interferon response, a phenomenon we named TRAIN (Transcription of Repeats Activates INterferon). Here, we report that curaxin, an anticancer small molecule, destabilizing nucleosomes via disruption of histone/DNA interactions, also induces TRAIN. Furthermore, curaxin inhibits oncogene-induced transformation and tumor growth in mice in an interferon-dependent manner, suggesting that anticancer activity of curaxin, previously attributed to p53-activation and NF-kappaB-inhibition, may also involve induction of interferon response to epigenetic derepression of the cellular 'repeatome'. Moreover, we observed that another type of drugs decondensing chromatin, HDAC inhibitor, also induces TRAIN. Thus, we proposed that TRAIN may be one of the mechanisms ensuring epigenetic integrity of mammalian cells via elimination of cells with desilenced chromatin.
DOI: https://doi.org/10.7554/eLife.30842.001

*For correspondence:
katerina.gurova@roswellpark.org

**Competing interests:** The authors declare that no competing interests exist.

## Introduction

Control of the integrity of genetic information in cells includes the activation of the DNA damage response, DNA-repair pathways and elimination of cells with damaged DNA (reviewed in [*Miller, 2010*; *Wang and Lindahl, 2016*]). The control of the integrity of epigenetic information is equally important and critical for the development and function of multicellular organisms, but far less studied. Epigenetic information is stored as chromatin, the highly organized complex of DNA, histone proteins and their chemical modifications (reviewed in [*Campos et al., 2014*; *Miska and Ferguson-Smith, 2016*]). Destabilization of chromatin should lead to the dysregulation of the cellular transcriptional program and loss of cell identity. One of the examples that demonstrate the high stability of the cellular epigenome is the well-known, extremely low efficiency of reprogramming of differentiated cells (reviewed [*Ebrahimi, 2015*; *Hussein and Nagy, 2012*]). Another example is the absence of one common 'cancer cell' phenotype: transcriptome analysis has clearly demonstrated that tumors, including cell lines propagated for years in culture, bear easily identifiable traits of the

tissue of origin in their transcriptional program (TCGA data). Nonetheless, we do not know what mechanisms control the stability of chromatin in cells, that is, whether there is any stress response to chromatin destabilization or negative selection against cells that lose epigenetic integrity.

One of the reasons for this deficit was the absence of tools to induce 'chromatin damage' without simultaneous introduction of DNA damage. This possibility appeared, when we found anticancer small molecules, known as curaxins, that were able to cause chromatin disassembly in cells in the absence of DNA damage (*Gasparian et al., 2011*), (*Safina et al., 2017*). These small molecules were identified in a phenotype-based screening for the ability to simultaneously activate p53 and inhibit NF-kappaB (*Gasparian et al., 2011*). Our search for the mechanism of action of curaxins resulted in the discovery that curaxins bind DNA via intercalation of the carbazole body accompanied by the protrusion of two side chains into the major groove and a third side chain into the minor groove of DNA. Curaxins have no effect on DNA chemical structure, that is, they do not cause any DNA damage in mammalian cells. However, their binding to DNA changes both the helical shape and flexibility of DNA and reduces its negative charge (*Safina et al., 2017*).

The structural unit of chromatin in eukaryotic cells is the nucleosome, which consists of a core of positively charged histone proteins wrapped in negatively charged DNA. The stability of the nucleosome is based on the electrostatic interactions between the histones and phosphate backbone of DNA, which are supported by several precise spatially-oriented points of contact between histone amino acids and the base pairs of DNA (*Luger et al., 1997*). When nucleosomal DNA is unwrapped, histone octamer becomes unstable due to the repulsion of the positively charged histones from each other. Alterations of DNA helical shape, charge and flexibility, caused by curaxins, force DNA to dissociate from the octamer in vitro and in vivo (*Safina et al., 2017*). At lower concentrations of curaxin CBL0137, DNA is first unfolded from the outer parts of the nucleosome, that is, from H2A/H2B dimers located at both sides of the inner H3/H4 tetramer, which leads to the detachment of the H2A/H2B dimer(s) and exposure of the surface of the H3/H4 tetramer. At higher CBL0137 concentrations, DNA is unwrapped from the inner H3/H4 tetramer, which leads to complete nucleosome disassembly (*Safina et al., 2017*). At genomic regions where the DNA helix loses histones, it becomes significantly under twisted, which in principle may be alleviated by topoisomerases or free rotation of DNA. However, in cells, free rotation is impossible due to the length of DNA and the binding to multiple proteins. Moreover, topoisomerases cannot cleave DNA in the presence of curaxins (*Safina et al., 2017*). Under these conditions, the excessive energy of negative supercoiling causes base unpairing and DNA transition from B-DNA into alternative non-B DNA structures. We have shown a massive transition of right-handed B-DNA in cells treated with curaxin CBL0137 into the left-handed Z-DNA form (*Safina et al., 2017*). Thus, treatment of cells with CBL0137 leads to dose-dependent destabilization and disassembly of chromatin in the absence of DNA damage and, most importantly, of the classical DNA damage response, providing an opportunity to study how a cell reacts to chromatin damage and the role of chromatin perturbations in the anticancer activity of curaxins.The first type of response that we observed in curaxin-treated cells was the activation of p53, a well-known reaction of cells to DNA damage. However, no phosphorylation of p53 N-terminal serines, which are obligatory markers of the DNA damage response, was detected in curaxin-treated cells. Furthermore, there was no activation of DNA-damage-sensitive kinases, such as ATM, ATR, and DNA-PK. Instead, p53 in curaxin-treated cells was phosphorylated at serine 392 by casein kinase 2 (CK2) that formed a complex with the histone chaperone Facilitates Chromatin Transcription (FACT) (*Gasparian et al., 2011*). FACT consists of two conservative proteins present in all eukaryotes, Suppressor of Ty 16 (SPT16) and Structure Specific Recognition Protein 1 (SSRP1). In basal conditions FACT is involved in transcription elongation (*Belotserkovskaya et al., 2003*), (Gurova and Studitsky, 2018; in preparation). Human FACT binds poorly intact nucleosomes (*Tsunaka et al., 2016*), (*Safina et al., 2017*) however, it may bind histone oligomers via different domains of SSRP1 and SPT16, when they become exposed during transcription (Gurova and Studitsky, 2018; in preparation). FACT binding prevents histone oligomers from dissociating while DNA is transcribed (*Belotserkovskaya et al., 2003*). We have found that nucleosome disassembly caused by curaxin opens multiple FACT-binding sites, which are normally hidden inside the nucleosome. Chromatin unfolding occurs first in heterochromatic regions, and all cellular FACT is rapidly relocated into these regions and depleted from the nucleoplasm (*Safina et al., 2017*). We named this phenomenon chromatin-trapping or c-trapping. Although curaxins do not bind and inhibit FACT directly, they cause exhaustion of FACT by trapping it in heterochromatin. Therefore, c-trapping is

equivalent to functional FACT inhibition. We also observed that FACT not only binds partially disassembled nucleosomes but also Z-DNA via the CID (c-terminal intrinsically disordered) domain of the SSRP1 subunit (*Safina et al., 2017*). Importantly, it was already known that the immediate neighboring domain, HMG (high mobility group), also binds non-B DNA, bent or cruciform (*Gariglio et al., 1997*; *Krohn et al., 2003*; *Yarnell et al., 2001*). In all cases of SSRP1 binding to DNA, there is a CK2-mediated signal for p53 activation (*Gasparian et al., 2011*; *Krohn et al., 2003*). Therefore, c-trapping may be part of the general response to the destabilization of chromatin and may serve to recruit FACT to the regions of potential nucleosome loss to prevent this loss and to restore chromatin structure.

However, there are many unclear issues within this model. Among the most immediate are two related questions: (i) what are the consequences, besides c-trapping, of nucleosome destabilization and chromatin unfolding in cells? (ii) Are all the effects of curaxins explained by the functional inactivation of FACT? In this study, we tried to answer these questions starting from the comparison of the effects of CBL0137 on transcription. We found that independently of the presence of FACT, CBL0137 activates transcription from heterochromatic regions normally silenced in cells, including centromeric and pericentromeric repeats and endogenous retroviral elements, associated with accumulation of double-stranded RNAs (dsRNA) in cells and activation of the type I interferon (IFN) response. This phenomenon, named TRAIN (Transcription of Repeats Activates INterferon) (*Leonova et al., 2013*) was observed previously upon DNA demethylation (*Chiappinelli et al., 2015*; *Guryanova et al., 2016*; *Roulois et al., 2015*). Although IFN activation has been traditionally viewed as part of an antiviral defense, TRAIN may present a more general cell response to the problems with chromatin organization in cells or a defense against loss of epigenetic integrity. We proposed that the reason for TRAIN in CBL0137-treated cells may be the direct effect of the drug on nucleosome structure leading to the opening of chromatin.

## Results

### Treatment with CBL0137 changes transcription of genes in FACT-positive and -negative tissues

The effect of CBL0137 on gene expression may be due to its binding to DNA and the unfolding of chromatin or due to the functional inhibition of FACT. To detect the potential FACT-independent effects of CBL0137, we sought to compare gene expression profiles in FACT-positive and -negative cells before and after CBL0137 treatment. All tumor and non-tumor cell lines tested to date express FACT subunits at different levels. The suppression of FACT expression with shRNA resulted in either partial reduction of FACT expression or complete knockdown of FACT in only a fraction of cells, which were quickly overgrown by FACT-positive cells (*Carter et al., 2015*; *Dermawan et al., 2016*; *Garcia et al., 2013*). To overcome this problem, we tested the effect of CBL0137 in vivo in tissues that either naturally express FACT or not. Liver and lung were selected as FACT-negative, and spleen and testis were used as FACT-positive tissues ([*Garcia et al., 2011*] and *Figure 1A*). Gene expression in these tissues was compared in mice treated with different doses of CBL0137 24 hr before organ collection. The three doses used represented different degrees of anticancer activity (absent, moderate and strong) with 90 mg/kg being close to the maximal tolerable dose (*Figure 1—figure supplement 1A*).

Hybridization analysis using mouse Illumina BeadChip array showed that all samples were clustered according to their tissue of origin and dose of CBL0137 (*Figure 1—figure supplement 1B*). The liver and spleen samples from the vehicle or 30 mg/kg CBL0137-treated mice were grouped together, suggesting little or no effect of this dose on gene expression in the tested organs (*Figure 1—figure supplement 1B*). Samples from mice treated with 60 and 90 mg/kg CBL0137 were also grouped together (spleen, testis) or close to each other (liver, lung), demonstrating a minimal difference between these doses. Surprisingly, very few genes changed expression in the testis (FACT-positive organ (*Figure 1—figure supplement 1C*), which may be due to either limited accumulation of the drug in testis as the result of the blood-testis barrier ('Sertoli cell barrier' (*Mruk and Cheng, 2015*) or the specific chromatin structure in most cells of this organ (*Wu and Chu, 2008*). Maximal changes were observed in the FACT-positive spleen followed by lung and liver (FACT-

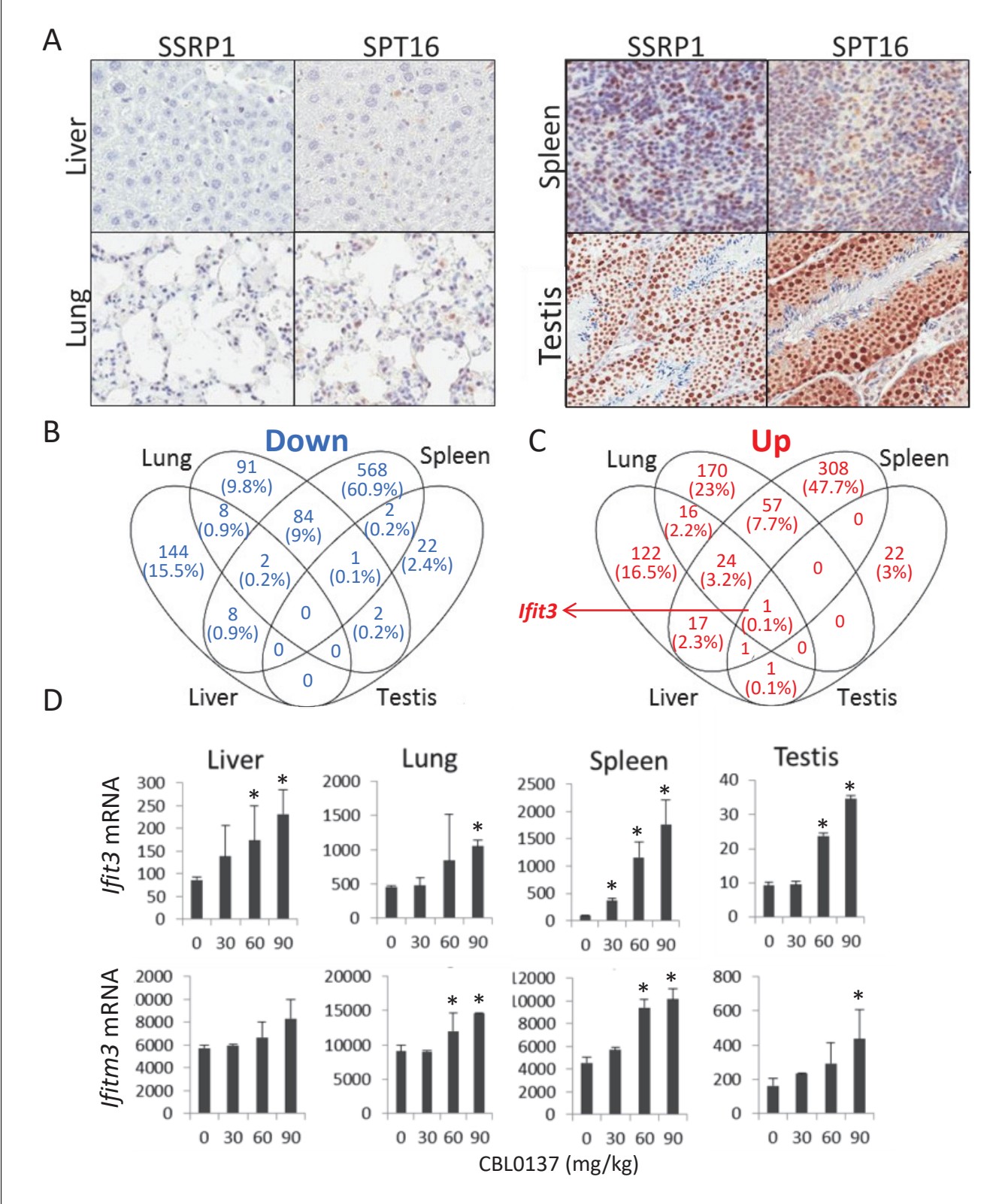

**Figure 1.** Effect of CBL0137 on transcription of genes in FACT-positive and -negative mouse tissues. (A) IHC staining with SSRP1 antibody of liver and lung tissues with no detectable SSRP1 and spleen and testis expressing SSRP1. (B, C) Analyses of changes in genes expression in different tissues of mice treated with 30, 60 or 90 mg/kg of CBL0137 using microarray hybridization. Venn diagrams of the lists of genes for which expression was downregulated (B) or upregulated (C) in response to CBL0137 in different organs in response to any of doses of CBL0137. (D) Dose-dependent

*Figure 1 continued on next page*

*Figure 1 continued*

upregulation of expression of *Ifit3* and *Ifitm3* genes in different organs. Mean normalized value of microarray hybridization signals of two biological replicates ± SD. Asterisks indicate conditions when expression was increased >1.5 folds with p-value<0.05.

DOI: https://doi.org/10.7554/eLife.30842.002

The following figure supplement is available for figure 1:

**Figure supplement 1.** Effects of different doses of CBL0137 in tumor growth and gene expression in mice.

DOI: https://doi.org/10.7554/eLife.30842.003

negative organs) (*Figure 1—figure supplement 1C*). The changes in gene expression caused by CBL0137 in these FACT-negative tissues suggest a FACT-independent mechanism.

There was almost no overlap in genes downregulated in response to CBL0137 among different organs (*Figure 1B*, *Figure 2—source data 1–6*). However, expression of one gene, *Ifit3*, was upregulated in all organs > 1.5-fold with p<0.05 (*Figure 1C,D*). Expression of another gene, *Ifit3m*, was upregulated in most of organs, but did not reach the formal criterion for upregulation, since its induction in liver was 1.45-fold (*Figure 1D*). Many transcripts in two or three organs increased similarly in a dose-dependent manner in response to CBL0137 (*Figure 1C*), suggesting that there may be a common pathway(s) induced by CBL0137 independently of FACT. To identify this pathway(s), we performed gene set enrichment analysis (GSEA) using the Molecular Signature Database (Broad Institute). The interferon (IFN) signaling pathway was upregulated in response to doses above 30 mg/kg in all organs except testis, which was not analyzed due to the low number of upregulated genes (*Figure 2A*, *Figure 2 – Figure Supplement 1* and *Figure 2—source data 1–6*). The same pathway was ranked first when the list of genes that were upregulated in response to CBL0137 in two or more organs was analyzed (*Figure 2B*). The transcription factors that regulate genes elevated in response to CBL0137 are STAT1, IRF1, IRF8 and IRF7 (*Figure 2C*). This analysis suggested that CBL0137 treatment leads to the activation of the IFN response in all tissues independently of FACT expression.

## CBL0137 causes rapid and robust IFN response in mouse tissues

We have shown previously that CBL0137 inhibits NF-kappaB via a FACT-dependent mechanism (*Gasparian et al., 2011*). Therefore, the activation of the IFN response was an unexpected finding, because the two pathways, NF-kappaB and IFN, have multiple common inducers and regulators (reviewed in [*Pfeffer, 2011*]). Hence, we confirmed the microarray data using different approaches. We observed an increase in the liver, lung, and spleen of the mRNA level of *Irf7*, a transcription factor whose expression is induced by IFN (*Wathelet et al., 1998*), following treatment with greater than 30 mg/kg CBL0137 (*Figure 3A* and *Figure 3—figure supplement 1A*). The same trend was observed in testis, but it did not reach statistical significance. Activation of the IFN response was confirmed using a second mouse strain by RT-PCR detection of an expanded set of IFN-inducible mRNAs, and at the protein level in spleen using an antibody to ISG49, the protein encoded by *Ifit3* gene (*Figure 3B,C*, and *Figure 3—figure supplement 1B*).

There are multiple known inducers of the IFN response, among which are components of viruses (e.g. dsRNA, cytoplasmic DNA), cytokines, DNA damage, and demethylation of genomic DNA (reviewed in [*Silin et al., 2009*]). Based on the literature, the kinetics of the IFN response are different depending on the inducer, with viral components and cytokines being the quickest (minutes – several hours [*Silin et al., 2009*]), followed by DNA damage (16–48 hr [*Brzostek-Racine et al., 2011*]), and demethylating agents (>48 hr [*Leonova et al., 2013*]). To compare the effect of CBL0137 vis-à-vis these stimuli, we assessed the kinetics of induction of the IFN response after CBL0137 treatment. We first measured *Irf7* mRNA in the spleen and lung of mice treated with three doses of CBL0137 for 24, 48, or 96 hr before organ collection. The peak of induction was already seen at 24 hr following treatment with either 60 or 90 mg/kg CBL0137, while 30 mg/kg showed slower kinetics (*Figure 3D,E*, and *Figure 3—figure supplement 2A,B*). Increased levels of two other IFN responsive transcripts, *Isg15* and *Usp18*, were detected as early as 6 hrspost-treatment with 90 mg/kg CBL0137 and reached a peak at 12 hr, then gradually declined (*Figure 3F,G*, and *Figure 3— figure supplement 2C*). Similar kinetics were observed at the protein level (*Figure 3H*). Thus, the

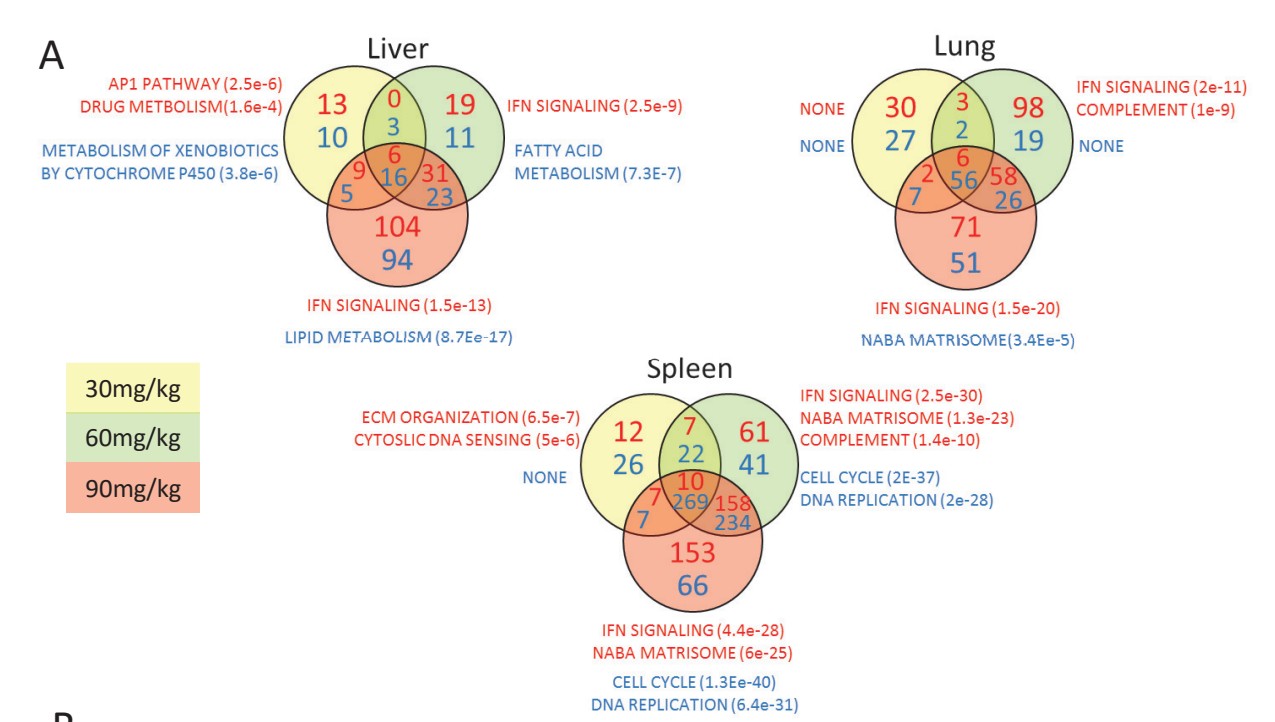

**B**

Gene sets enriched for genes upregulated by CBL0137 in any two of organs

| Gene Set Name | No Genes in Gene Set (K) | No Genes in Overlap (k) | k/K | P-value | FDR q-value |
|---|---|---|---|---|---|
| REACTOME_INTERFERON_ALPHA_BETA_SIGNALING | 64 | 14 | 0.2188 | 1.44E-26 | 1.91E-23 |
| REACTOME_INTERFERON_SIGNALING | 159 | 16 | 0.1006 | 1.76E-24 | 1.17E-21 |
| REACTOME_CYTOKINE_SIGNALING_IN_IMMUNE_SYSTEM | 270 | 17 | 0.063 | 2.00E-22 | 8.84E-20 |
| REACTOME_IMMUNE_SYSTEM | 933 | 23 | 0.0247 | 5.60E-21 | 1.86E-18 |
| REACTOME_INTERFERON_GAMMA_SIGNALING | 63 | 6 | 0.0952 | 1.05E-09 | 2.80E-07 |
| REACTOME_REGULATION_OF_IFNA_SIGNALING | 24 | 4 | 0.1667 | 7.15E-08 | 1.58E-05 |
| ST_TYPE_I_INTERFERON_PATHWAY | 9 | 3 | 0.3333 | 3.62E-07 | 5.81E-05 |
| REACTOME_INNATE_IMMUNE_SYSTEM | 279 | 7 | 0.0251 | 4.31E-07 | 5.81E-05 |
| PID_IL23_PATHWAY | 37 | 4 | 0.1081 | 4.38E-07 | 5.81E-05 |

**C**

Lists of transcriptional factors, which targets were upregulated by CBL0137 in any of organs

| Gene Set Name | No Genes in Gene Set (K) | Description | # Genes in Overlap (k) | k/K | P-value | FDR q-value |
|---|---|---|---|---|---|---|
| V$ISRE_01 | 247 | STAT1 | 10 | 0.0405 | 1.17E-11 | 7.21E-09 |
| STTTCRNTTT_V$IRF_Q6 | 188 | Unknown | 8 | 0.0426 | 1.00E-09 | 3.09E-07 |
| V$IRF_Q6 | 242 | IRF1 | 8 | 0.0331 | 7.29E-09 | 1.36E-06 |
| V$ICSBP_Q6 | 248 | IRF8 | 8 | 0.0323 | 8.83E-09 | 1.36E-06 |
| V$IRF7_01 | 252 | IRF7 | 6 | 0.0238 | 4.06E-06 | 5.00E-04 |
| TGANTCA_V$AP1_C | 1121 | JUN | 9 | 0.008 | 9.76E-05 | 1.00E-02 |
| YATGNWAAT_V$OCT_C | 360 | Uknown | 5 | 0.0139 | 3.36E-04 | 2.95E-02 |

**Figure 2.** Activation of IFN response is detected in response to CBL0137 in different tissues of mice via GSEA. (**A**) Venn diagrams showing the number of up (red font) or down (blue font) regulated genes in response to CBL0137 together with gene set names enriched for these genes. The p-value of overlap is shown in parentheses. (**B**) GSEA of curated signaling pathways using the list of genes upregulated in two or more tissues in response to any

*Figure 2 continued on next page*

*Figure 2 continued*

dose of CBL0137. (**C**) GSEA of targets of known transcriptional factors using the list of genes upregulated in any of tissues in response to any of doses of CBL0137.

DOI: https://doi.org/10.7554/eLife.30842.004

The following source data and figure supplement are available for figure 2:

**Source data 1.** GSEA analysis of genes downregulated in response to treatment with 30 mg/kg (sheet 1), 60 mg/kg (sheet 2) and 90 mg/kg (sheet 3) of CBL0137 in liver.
DOI: https://doi.org/10.7554/eLife.30842.006
**Source data 2.** GSEA analysis of genes upregulated in response to treatment with 30 mg/kg (sheet 1), 60 mg/kg (sheet 2) and 90 mg/kg (sheet 3) of CBL0137 in liver.
DOI: https://doi.org/10.7554/eLife.30842.007
**Source data 3.** GSEA analysis of genes downregulated in response to treatment with 30 mg/kg (sheet 1), 60 mg/kg (sheet 2) and 90 mg/kg (sheet 3) of CBL0137 in lung.
DOI: https://doi.org/10.7554/eLife.30842.008
**Source data 4.** GSEA analysis of genes upregulated in response to treatment with 30 mg/kg (sheet 1), 60 mg/kg (sheet 2) and 90 mg/kg (sheet 3) of CBL0137 in lung.
DOI: https://doi.org/10.7554/eLife.30842.009
**Source data 5.** GSEA analysis of genes downregulated in response to treatment with 30 mg/kg (sheet 1), 60 mg/kg (sheet 2) and 90 mg/kg (sheet 3) of CBL0137 in spleen.
DOI: https://doi.org/10.7554/eLife.30842.010
**Source data 6.** GSEA analysis of genes upregulated in response to treatment with 30 mg/kg (sheet 1), 60 mg/kg (sheet 2) and 90 mg/kg (sheet 3) of CBL0137 in spleen.
DOI: https://doi.org/10.7554/eLife.30842.011
**Figure supplement 1.** GSEA analysis of genes upregulated in liver (**A**), lung (**B**) and spleen (**C**) of mice treated with CBL0137.
DOI: https://doi.org/10.7554/eLife.30842.005

CBL0137-induced IFN response is faster than that reported with other small molecules and is closer to the response observed with biologicals.

## Increased expression of IFN-responsive genes induced by CBL0137 is dependent on IFN signaling

Experiments with mice demonstrated that CBL0137 caused a rapid and robust induction of the IFN response in all tissues of mice that were tested. To investigate the mechanism of this phenomenon, we modeled this effect in vitro using normal and tumor cells (mouse and human). We detected an increase in mRNA and protein levels of IFN-inducible genes in both human and mouse fibroblasts (*Figure 4A–C*). Effect was stronger on p53 null, than in p53 wild-type MEF (*Figure 4B,C*), as we observed before with DNA demethylating agents (*Leonova et al., 2013*). However, no induction of IFN-inducible genes was detected in tumor cells of mouse or human origin. Since different components of the IFN response are frequently inactivated in tumors (*Kulaeva et al., 2003*), we employed a reporter assay that detected an IFN response through the activation of a consensus IFN-sensitive response element (ISRE), which drove the expression of mCherry or luciferase (*Imam et al., 1990*). We found that this way of assessment of IFN response is more sensitive, since in some tumor cells (e.g. HCT116) even treatment with IFN alpha did not cause induction of *IFIT3*, while ISRE reporter activity was easily detected (*Figure 4—figure supplement 1A,B*). Using these assays, we observed that CBL0137 also induces an IFN response in both mouse and human tumor cells (*Figure 4D,E* and *Figure 4—figure supplement 1*).

The next step toward the mechanism of CBL0137-activated IFN response was to establish whether this was, in fact, a response to IFNs, that is, was there an increase in the IFNs level after CBL0137 treatment. However, our attempts to detect IFNs (alpha, beta or gamma) in the plasma of mice treated with CBL0137 or in cell culture medium failed, which may be due to the short half-life, low levels of IFNs even after induction, or both. To overcome this limitation and to test whether the induction of IFN responsive transcription after CBL0137 treatment was due to IFNs, we used cells with different components of the IFN signaling pathway inactivated, such as MEF from mice with a knockout of either the type I IFN receptor (via *Ifnar1*) or *Irf1*, or MEF with shRNA to *Irf7* (transcription factors controlling the expression of IFN-responsive genes in response to IFN). We also used *Trp53*

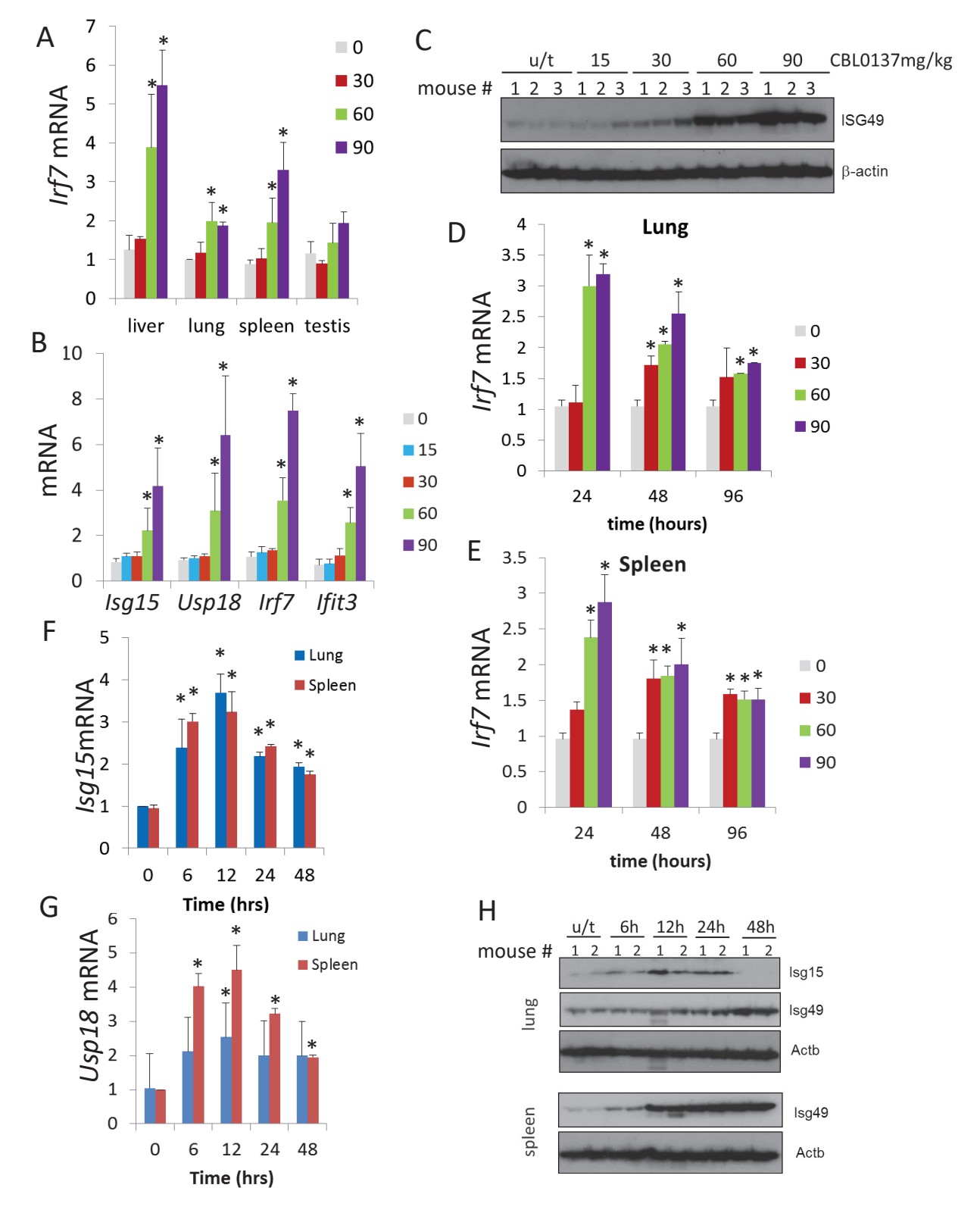

**Figure 3.** CBL0137 causes increased expression of IFN-responsive genes in different tissues of mice. Quantitation of RT-PCR data (**A, B, D, E, F, G**) shown as fold change upon treatment with different doses of CBL0137 (mg/kg) comparing to vehicle-treated control. Mean values from three mice ± SD. Immunoblotting of mouse plasma (**C**) or tissue lysates (**H**). (**A**) Treatment of C57Bl/6 mice for 24 hr. B and C. Treatment of NIH Swiss mice

*Figure 3 continued on next page*

*Figure 3 continued*

for 24 hr. D - H. Different time treatment of C57Bl/6 mice. C and H – numbers indicate individual mice in each group. Bars – mean of two or three replicates +SD, asterisk – p<0.05 vs untreated control.

DOI: https://doi.org/10.7554/eLife.30842.012

The following figure supplements are available for figure 3:

**Figure supplement 1.** Images of RT-PCR reactions used for quantitation on *Figure 3*.

DOI: https://doi.org/10.7554/eLife.30842.013

**Figure supplement 2.** Images of RT-PCR reactions used for quantitation for *Figure 3*.

DOI: https://doi.org/10.7554/eLife.30842.014

-/- MEF since we have shown previously that the absence of p53 increases IFN signaling in response to dsRNA (*Leonova et al., 2013*). A deficiency in either *Ifnar1* and *Irf7* reduced or eliminated the induction of *Ifit3* expression after CBL0137, respectively. In contrast, the loss of p53 stimulated the response. There was no effect of *Irf1* knockout (*Figure 5A*). Similar results were obtained in vivo using spleen from *Ifnar*1-/-, *Irf7*-/- or *Trp53*-/- mice treated with CBL0137 (*Figure 5B*).

## CBL0137 treatment leads to transcription of repetitive heterochromatic genomic regions

A known trigger of the IFN response is a viral infection. There are several ways by which cells can detect viral invasion. First, viral dsRNA can be detected by several intra-cellular receptors, such as TLR3, 7 or 8, RIG1, and MDA5 (reviewed in [*Unterholzner, 2013*]). Additionally, the presence of DNA in the cytoplasm or non-B DNA, such as left-handed Z-DNA, found in some viruses (*Nordheim and Rich, 1983*) can also be recognized by a cell as a viral infection. In the latter case, the IFN response is activated via Z-DNA-binding protein 1 (ZBP1), also known as the DNA-dependent activator of IFN (DAI) (*Takaoka et al., 2007*). Importantly, we have recently shown that high doses of CBL0137 caused disassembly of nucleosomes in cells, which led to the extensive negative supercoiling of DNA and a transition from B-DNA into left-handed Z-DNA (*Safina et al., 2017*). Although this effect was observed at significantly higher CBL0137 concentrations (>2.5 μM) than that which induced an IFN response, we investigated whether the induction of Z-DNA transition by CBL0137 could be responsible for its effect on the IFN response. To do this, we deleted the *ZBP1* gene from HeLa and HT1080 cells using CRISP/Cas9 recombinase and two human ZBP1 gRNAs. The loss of *ZBP1* expression was confirmed by RT-PCR (*Figure 6—figure supplement 1A*). We compared the effect of CBL0137 on ISRE-mCherry reporter activity in these cells to wild-type cells and cells transduced with control gRNA and observed insignificant changes in the induction of the reporter activity 48 hr after CBL0137 treatment (*Figure 6—figure supplement 1B*). Importantly, there was no Z-DNA present in the cells treated with CBL0137 doses that caused a robust IFN response as judged by the absence of staining with Z-DNA antibodies (*Figure 6—figure supplement 1C*). Thus, we concluded that Z-DNA transition induced by CBL0137 does not play role in the induction of IFN response in CBL0137-treated cells.

We next determined whether dsRNA induced an IFN response in CBL0137-treated cells. Using an antibody to dsRNA (*Targett-Adams et al., 2008*), we observed increased staining in CBL0137-treated cells (*Figure 6A,B*). One potential endogenous source of dsRNA may be from the transcription of repetitive elements, such as endogenous retroviruses, centromeric, or pericentromeric repeats (*Leonova et al., 2013*). Transcription of these elements may be enhanced as a result of nucleosome destabilization and chromatin unfolding caused by CBL0137. Short Interspersed Nuclear Element (SINE) retrotransposons constitute one of the main components of the genomic repetitive fraction (*Kramerov and Vassetzky, 2011*). Hence, we first tested whether transcription of B1 SINE is increased upon CBL0137 treatment using northern blotting. In samples of untreated MEFs, binding of the B1 probe was detected for a number of different length transcripts, possibly due to the presence of B1 repeats within the introns of multiple mRNAs. After CBL0137 treatment, a single lower molecular weight band appeared that was strongly bound by the B1 probe and was similar in size and abundance to the band in cells treated with the positive control (5-aza-cytidine (5-aza)) (*Figure 6C*). To determine whether other transcripts that can form dsRNAs appear in cells treated with CBL0137, we performed RNA-sequencing (RNA-seq) analysis using total RNA from wild-type

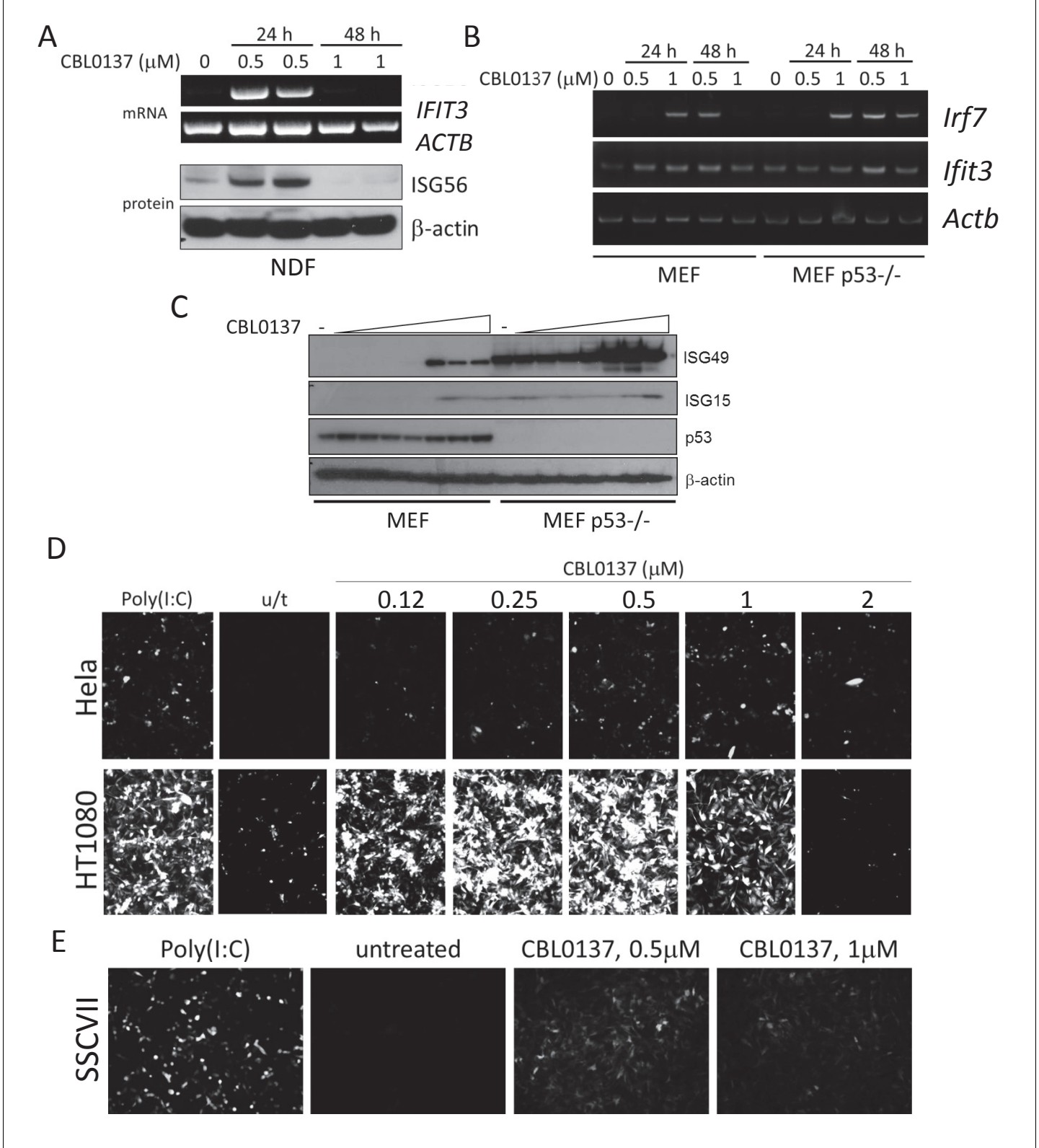

**Figure 4.** Induction of IFN response in human and mouse cells treated with CBL0137. (**A**) RT-PCR and immunoblotting of mRNA and lysates from human normal diploid fibroblasts (NDF). ISG56 is a product of the human *IFIT3* gene. (**B**) RT-PCR of mRNA from MEF, wild type or *Trp53* (p53) deficient. (**C**) Immunoblotting of lysate of MEF treated with different doses of CBL0137 (0.03–2 µM) for 24 hr. D and E. Microphotographs of human (**D**) or mouse

*Figure 4 continued on next page*

*Figure 4 continued*

(E) tumor cells expressing ISRE-mCherry reporter, treated with different doses of CBL0137 for 48 hr. Poly(I:C) - polyinosinic:polycytidylic acid was used as a positive control for reporter activation at 50 µg/ml.

DOI: https://doi.org/10.7554/eLife.30842.015

The following figure supplement is available for figure 4:

**Figure supplement 1.** Comparison of the sensitivity of ISRE reporter assay with detection of the transcript of endogenous ISG, *IFIT3* in human tumor cell line HCT116 transduced with lentiviral ISRE-Luc reporter.

DOI: https://doi.org/10.7554/eLife.30842.016

and *Trp53-/-* MEF treated with CBL0137 in vitro or lungs from mice collected 24 hr after treatment with 60 mg/kg CBL0137 in vivo. This analysis demonstrated a significant increase in the number of transcripts corresponding to several classes of endogenous viruses as well as GSAT (pericentromeric) and SATMIN (centromeric) repeats in cells (*Figure 6D*, *Figure 6—figure supplements 1* and *Figure 6—source data 1*). These data suggest that treatment with CBL0137 leads to the activation of the transcription of heterochromatic regions, which are not or minimally transcribed under basal conditions.

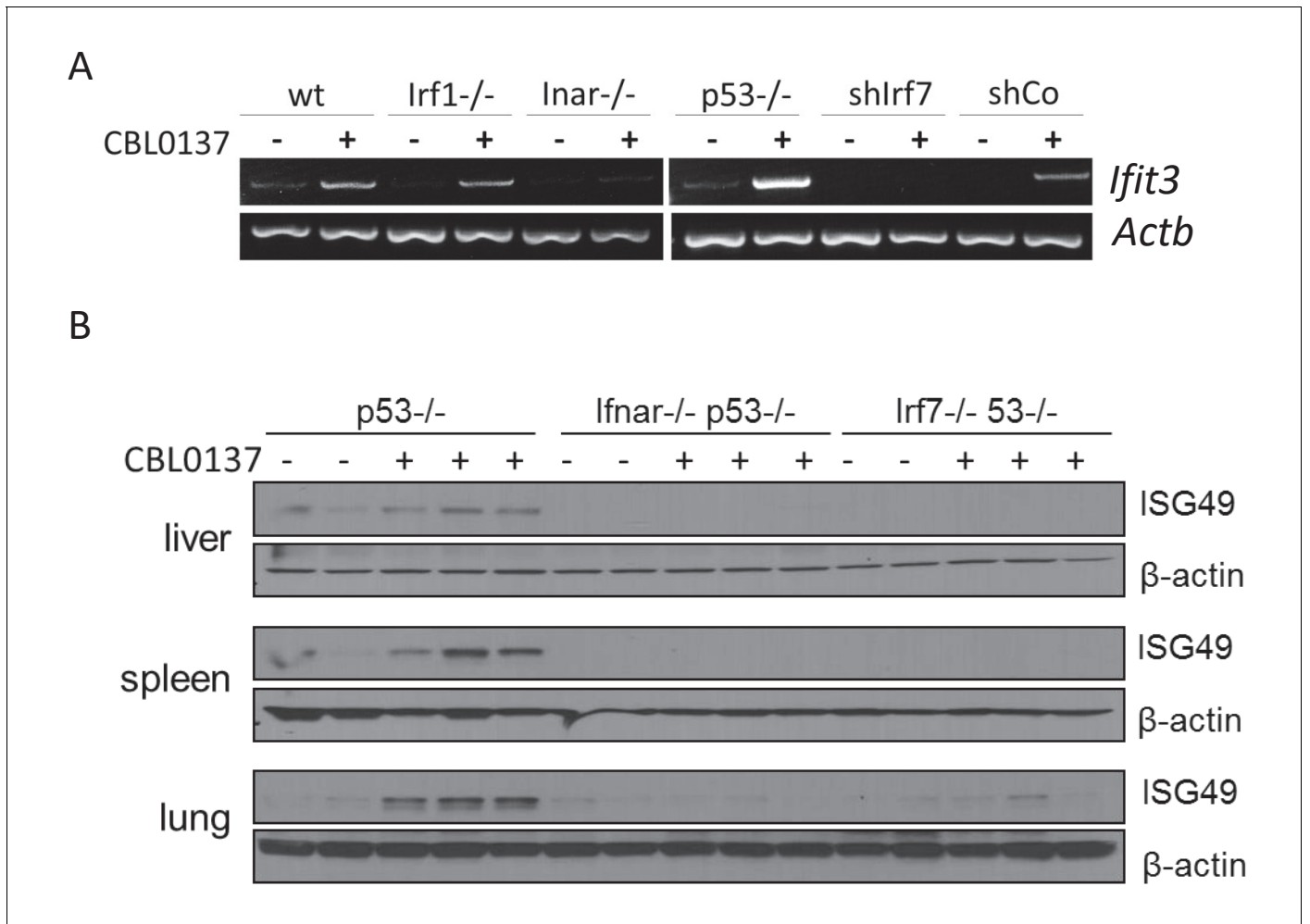

**Figure 5.** Induction of IFN-responsive gene by CBL0137 depends on IFN signaling. (**A**) RT-PCR of RNA from MEF cells of different genotype or wild type MEF transduced with control shRNA or shRNA to *Irf7* and treated for 24 hr with 0.5 µM of CBL0137. (**B**) Immunoblotting of tissue lysates from mice of different genotypes treated with vehicle control (n = 2 for each genotype) or 60 mg/kg CBL0137 (n = 3) 24 hr before tissue collection.

DOI: https://doi.org/10.7554/eLife.30842.017

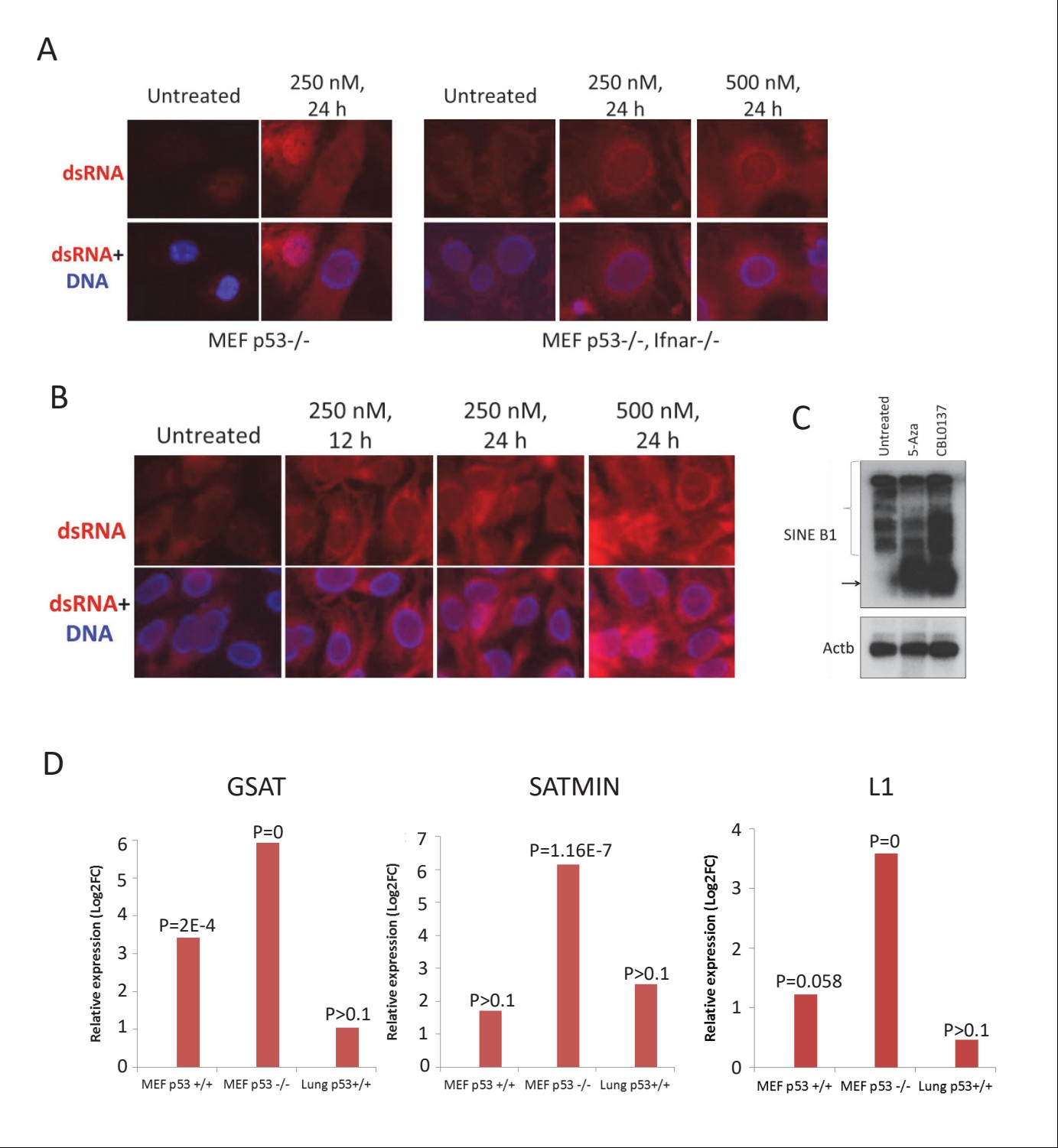

**Figure 6.** Elevated presence of double-stranded RNA in cells and tissues of mice treated with CBL0137. Immunofluorescent staining of MEF (**A**) and human HT1080 (**B**) cells with antibody to dsRNA (red) and Hoeschst (blue). (**C**) Northern blotting of RNA from MEF (*Trp53-/-*) treated with either 5-Aza for 72 hr or CBL0137 for 48 hr and hybridized with B1 or *Actb* probes. (**D**) Increase in transcripts corresponding to major (GSAT) or minor (SATMIN) satellites or LINE1 in MEF cells or lung tissue of mice treated with either 1 µM or 60 mg/kg of CBL0137 for 24 hr. Bars are relative increase of the transcripts abundance in treated versus untreated samples. p=FDR corrected p-value (see Materials and methods for details).

DOI: https://doi.org/10.7554/eLife.30842.018

The following source data and figure supplement are available for figure 6:

*Figure 6 continued on next page*

*Figure 6 continued*

**Source data 1.** Analysis of expression of repetitive elements in control and CBL0137 treated wild type of *Trp53-/-* MEF cells of lungs of wild-type mice via RNA-seq.

DOI: https://doi.org/10.7554/eLife.30842.020

**Figure supplement 1.** INF induction by CBL0137 does not depend on ZBP1.

DOI: https://doi.org/10.7554/eLife.30842.019

## Opening of chromatin is accompanied by TRAIN

CBL0137 may induce transcription of genomic regions that are silenced at physiological conditions via several different mechanisms. Loss of FACT via c-trapping may lead to activation of silenced transcription since mutations of FACT subunits in yeast causes transcription from cryptic sites that are covered by nucleosomes in basal conditions (*Mason and Struhl, 2003*), (*Jamai et al., 2009*). FACT is also important for chromatin assembly at centromeres (*Okada et al., 2009*; *Prendergast et al., 2016*). We observed the induction of the IFN response in organs that lacked FACT (lung and liver). However, the absence of FACT in these tissues does not exclude the role of FACT in the prevention of the response. Nevertheless, it was clear that there are FACT-independent effects of CBL0137 on transcription.

Opening or disassembly of nucleosomes due to CBL0137 binding to DNA may increase accessibility of DNA to RNA Polymerase (RNAP) II. To see whether CBL0137 can induce transcription of a reporter silenced due to chromatin condensation, we used HeLa-TI cells (*Poleshko et al., 2010*; *Shalginskikh et al., 2013*), which contain an integrated avian sarcoma virus genome with a silenced green fluorescent protein (GFP) gene. Under basal conditions, GFP is expressed in very few cells. However, the opening of chromatin using the HDAC inhibitor trichostatin A (TSA) activates GFP expression in a significant proportion of cells (*Figure 7A*). An increase in the proportion of GFP-expressing cells was also observed after CBL0137 treatment (*Figure 7A*).

Next, we tested whether chromatin opening by HDAC inhibition will be accompanied by TRAIN. We treated MEF and HeLa-TI cells with TSA or CBL0137 and monitored the activation of IFN signaling by RT-PCR in MEF or ISRE-mCherry reporter in Hela-TI cells (*Figure 7B,C*). In MEF cells, both TSA and CBL0137 induced expression of *Ifit3* and *Irf7* genes (*Figure 7B*), while in HeLa-TI cells, the robust induction of ISRE reporter was evident only after CBL0137 treatment. TSA caused only a weak increase in ISRE reporter activity (1.5 folds versus >5 folds for CBL0137) only at the highest concentration (500 nM, *Figure 7C*) when evidence of toxicity was present. At the same time, the degree of activation of the silenced GFP in HeLa-TI cells was similar for TSA and CBL0137 (*Figure 7C*).

Importantly, although GFP expression was increased by both agents, the mechanisms of chromatin opening were different for TSA and CBL0137. As expected, TSA treatment caused an increase in acetylated histone H3 levels, while CBL0137 decreased the amount of chromatin-bound histone H3 (*Figure 7D*). However, as we saw previously (*Safina et al., 2017*), substantial loss of core histones (i.e. chromatin disassembly) was observed with approximately 5 µM CBL0137, which is much higher than the concentrations at which we observed TRAIN (0.5–1 µM). We proposed that nucleosome 'opening,' that is, the state of loose contact between the DNA and histone core without loss of histones from the core (proposed in the review [*Zlatanova et al., 2009*]), may be enough to create conditions for TRAIN. Namely, no nucleosome disassembly or dissociation of nucleosome parts is needed. Although there is no clear definition of an opened and a closed nucleosome state, the closed nucleosome in cells is additionally 'locked' by the linker histone H1 (*Syed et al., 2010*; *Zhou et al., 2013*). Thus, we investigated how the distribution of H1 is changed upon treatment with CBL0137 using imaging of live cells that express m-Cherry-tagged histone H1.5. CBL0137 treatment led to a redistribution of H1.5 from the chromatin to the nucleoli, similarly to the effect of CBL0137 on core histones (*Safina et al., 2017*) but this phenomenon was observed at a faster and at much lower CBL0137 concentrations (*Figure 8A,B*). TRAIN in CBL0137-treated cells was observed at the same concentration range as redistribution of histone H1 and before the disassembly of the nucleosome core, suggesting that nucleosome opening is sufficient to allow the transcription of heterochromatic regions in cells. Thus, we observed the presence of TRAIN using different chromatin modifying agents; however, its degree depends on cell type and the mode of chromatin opening.

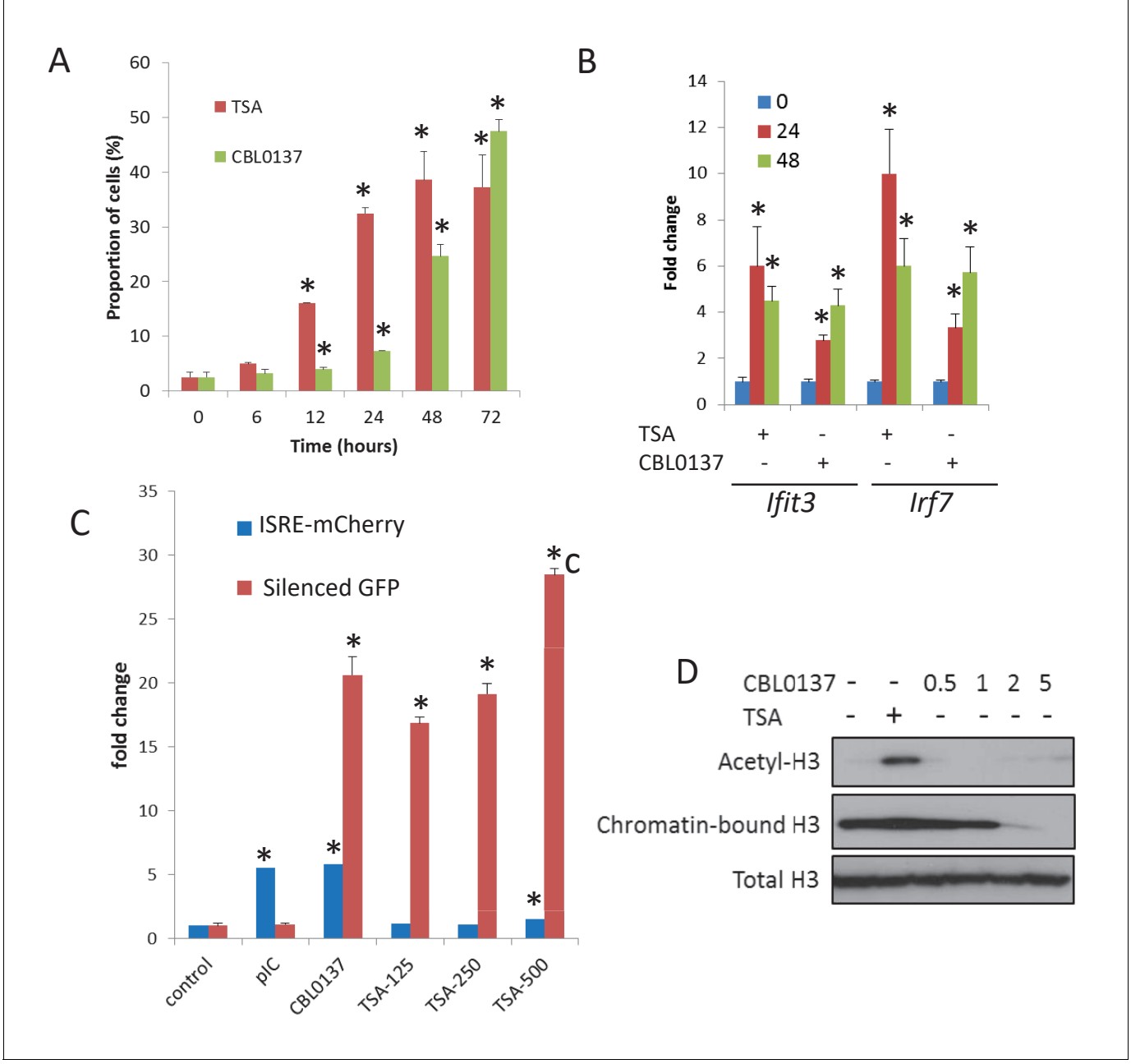

**Figure 7.** Chromatin opening is accompanied by TRAIN. (**A**) Increase in the proportion of cells expressing GFP from silenced viral promoter in Hela cells treated with either TSA (250 nM) or CBL0137 (500 nM). (**B**) Changes in the abundance of *Ifit3* and *Irf7* mRNAs in MEF treated with 500 nm of TSA or CBL0137 for 24 or 48 hr. Quantitation of RT-PCR, mean of two experiments ± SD. All changes are significant versus untreated control ($p < 0.05$). (**C**) Change in the proportion of Hela-TI-ISRE-mCherry cells positive for mCherry and GFP after 48 hr treatment with 10 µg/ml of polyI-C (pIC), 500 nM of CBL0137 or 125–500 nM of TSA. (**D**) Immunoblotting of extracts of Hela cells treated with TSA (200 nM) or different doses of CBL0137 (µM) for 24 hr. Acetyl H3 and total H3 were detected in total nuclear extracts, chromatin bound H3 - in chromatin pellet, washed from nucleoplasm with 450 mM NaCl buffer. Bars are mean of two replicates ± SD. Asterisk - $p < 0.05$ vs untreated control.

DOI: https://doi.org/10.7554/eLife.30842.021

The following figure supplement is available for figure 7:

**Figure supplement 1.** Abundance of transcripts corresponding to major (GSAT) or minor (SATMIN) satellites or LINE1 per millions of reads of RNA-seq of total RNA from MEF cells from *Trp53* wild type (p53+/+) or *Trp53*-null (p53-/-) mice or lung tissue of *Trp53* wild-type mice treated with either 1 µM or 60 mg/kg of CBL0137 for 24 hr.

DOI: https://doi.org/10.7554/eLife.30842.022

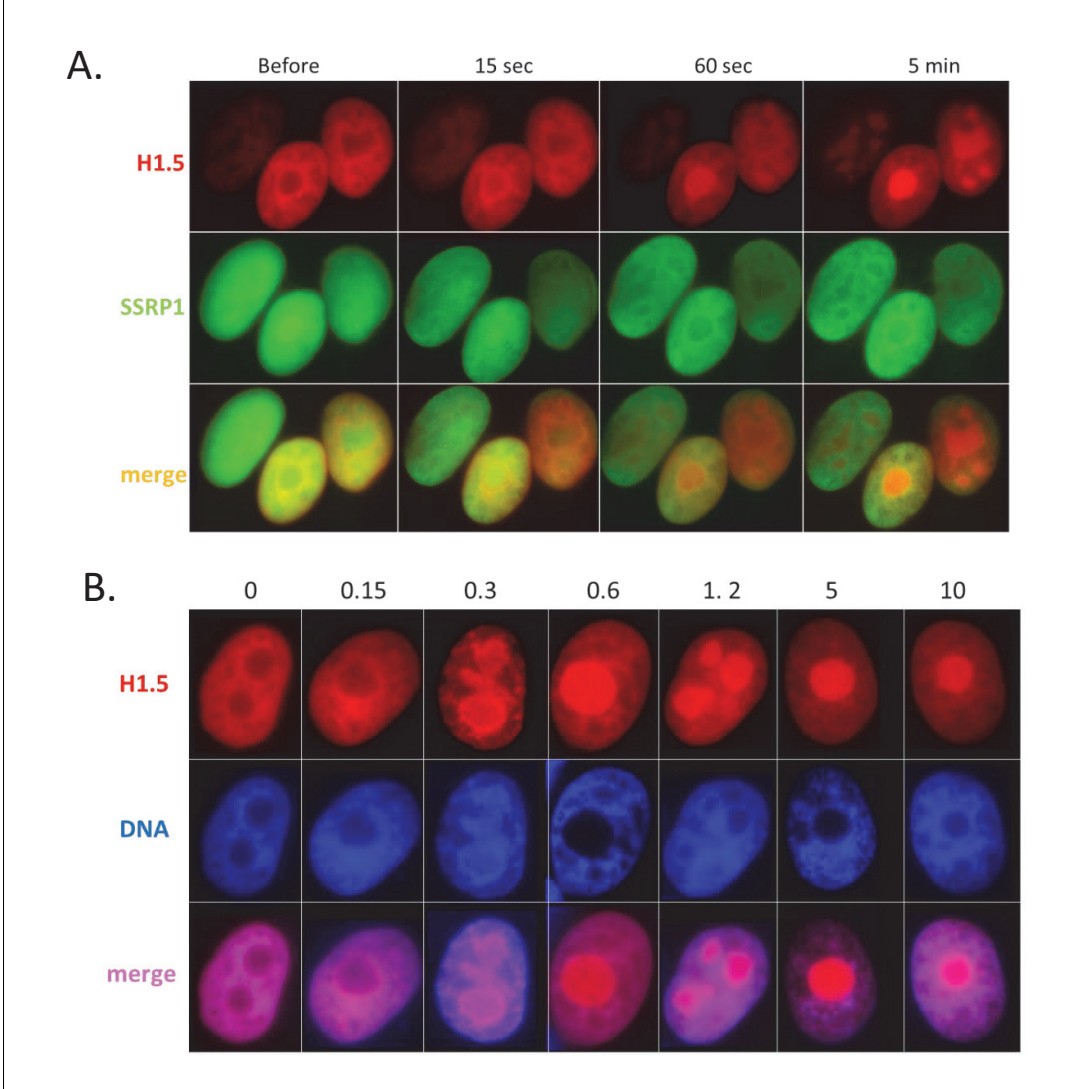

**Figure 8.** CBL0137 treatment leads to displacement of histone H1.  Live cell imaging of HT1080 cells expressing mCherry-tagged H1.5 and GFP-tagged SSRP1. (**A**) Images of nuclei of the same cells at different moments after addition of 1 µM CBL0137 to cell culture medium. (**B**) Images of nuclei of cells treated with different concentrations of CBL0137 (µM) for 1 hr.
DOI: https://doi.org/10.7554/eLife.30842.023

## TRAIN induced by CBL0137 impacts anti-cancer activity of CBL0137

We have previously shown that administration of CBL0137 to tumor prone mice prior to tumor onset significantly reduced the incidence of cancer and diminished the occurrence of aggressive forms of breast cancer (*Koman et al., 2012*). We explained this effect by the fact that CBL0137 inhibited NF-kappaB and activated p53, both well-established mechanisms of cancer prevention (rev. in [*Athar et al., 2011*] and [*Lin et al., 2010*]). However, activation of IFN may be an additional mechanism to eliminate transformed cells, because they are more sensitive to IFN-mediated apoptosis than normal cells (reviewed in [*Parker et al., 2016*]). To test this hypothesis, we used p53-negative MEF (to eliminate the effect of p53 activation) with a functional or disabled IFN response via knock-out of *Ifnar1* (*Picaud et al., 2002*). MEF cells have almost no basal NF-kappaB activity. Both cell types were transduced with a lentiviral construct for the expression of the mutant H-Ras$^{V12}$ oncogene. Forty-eight hours after lentiviral transduction, cells were treated with a low dose of CBL0137 for an additional 48 hr and then left to grow in drug-free medium. After 10 days, the number of

transformed foci was significantly reduced in cells treated with CBL0137 compared to untreated cells, but only if the cells had functional IFN receptor (*Ifnar1+/+*, *Figure 9A*). Thus, TRAIN may be an additional mechanism by which CBL0137 exerts its tumor preventive effect.

To evaluate the impact of TRAIN in the anticancer activity of CBL0137, we inoculated H-Ras[V12]-transformed MEF, wild type or *Ifanr1-/-*, into *Ifanr1-/-* mice. When tumors reached the size of 50 mm$^3$, we treated these mice with CBL0137 once a week for 3 weeks. Wild type tumors did not grow at all since the start of treatment, while *Ifanr1-/-* tumors continued growing, albeit slower than vehicle treated tumors (*Figure 9B,C*). Thus, induction of IFN response by CBL0137 is an important component of the anticancer activity of CBL0137.

## Discussion

Our study describes the consequences of chromatin destabilization caused by small molecule binding to DNA. CBL0137 was identified in a cell-based screen, and some of its effects, such as activation of p53 and inhibition of NF-kappaB, functional inactivation of FACT, and higher toxicity to tumor than normal cells, were already described (*Gasparian et al., 2011*), (*Barone et al., 2017*; *Carter et al., 2015*; *Dermawan et al., 2016*). However, recently, we demonstrated that binding of CBL0137 to DNA destabilizes nucleosomes in vitro and in vivo, leading to histone eviction and chromatin decondensation in cells (*Safina et al., 2017*). Since CBL0137 does not cause chemical DNA modifications and has no effect on transcription and replication of naked DNA in vitro at the concentrations used in this study, this creates a unique situation to assess how 'chromatin damage' influences transcription.

First, we would like to define here 'chromatin damage' as a change in the original chromatin composition that is artificially induced by a drug or other experimental manipulations. In the case of CBL0137, this occurs most probably in a few continuous phases (*Figure 10*). At lower concentrations of CBL0137 (<1 µM in cells), the nucleosome 'opens,' loses the linker histone H1 without dissociation or loss of core histones. At higher CBL0137 concentrations (1–5 µM), dimer detachment occurs with occasional loss of core histones. We clearly saw this stage in vitro as the appearance of hexasomes (*Safina et al., 2017*). At CBL0137 concentrations greater than 5 µM, the nucleosome is disassembled, and core histones are evicted from the chromatin. This sequence of events is in line with the reversibility of the toxic effects of CBL0137. Indeed, after a short incubation with <1 µM

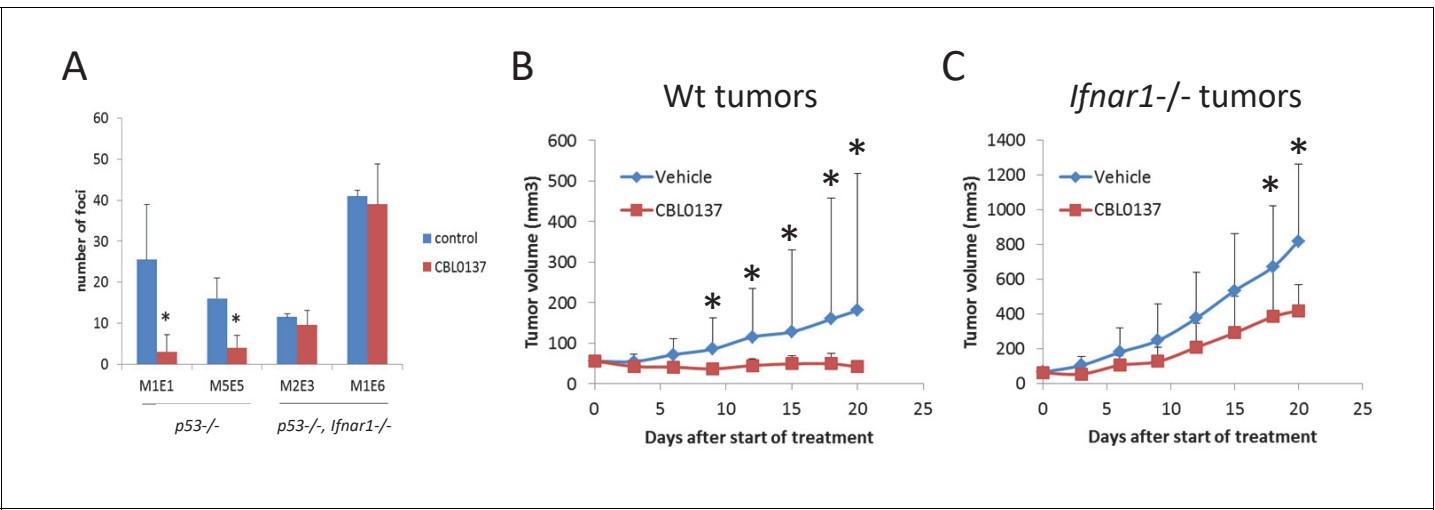

**Figure 9.** Input of IFN response into anti-cancer activity of CBL0137. (**A**) Treatment of cells transduced with H-Ras[V12] with CBL0137 leads to the reduction in the number of transformed foci in Ifnar-dependent manner. MEF of different background were transduced with lentiviral Hras[V12] followed by treatment with 250 nM CBL0137 for 48 hr. Number of foci was detected on day 10 after transduction. Cultures from two embryos of each genotype were used. Mean of two replicates ± SD. * - p<0.05. B-C. Effect of CBL0137 on the growth of wild type (wt) (**B**) of *Ifnar1-/-* (**C**) transformed MEF in *Ifnar1-/-* mice. Group of five mice of each genotype were treated with vehicle of 70 mg/kg of CBL0137 once a week for 3 weeks. Mean tumor volume + SD. Asterisk – p<0.05 between CBL0137 and vehicle-treated groups.
DOI: https://doi.org/10.7554/eLife.30842.024

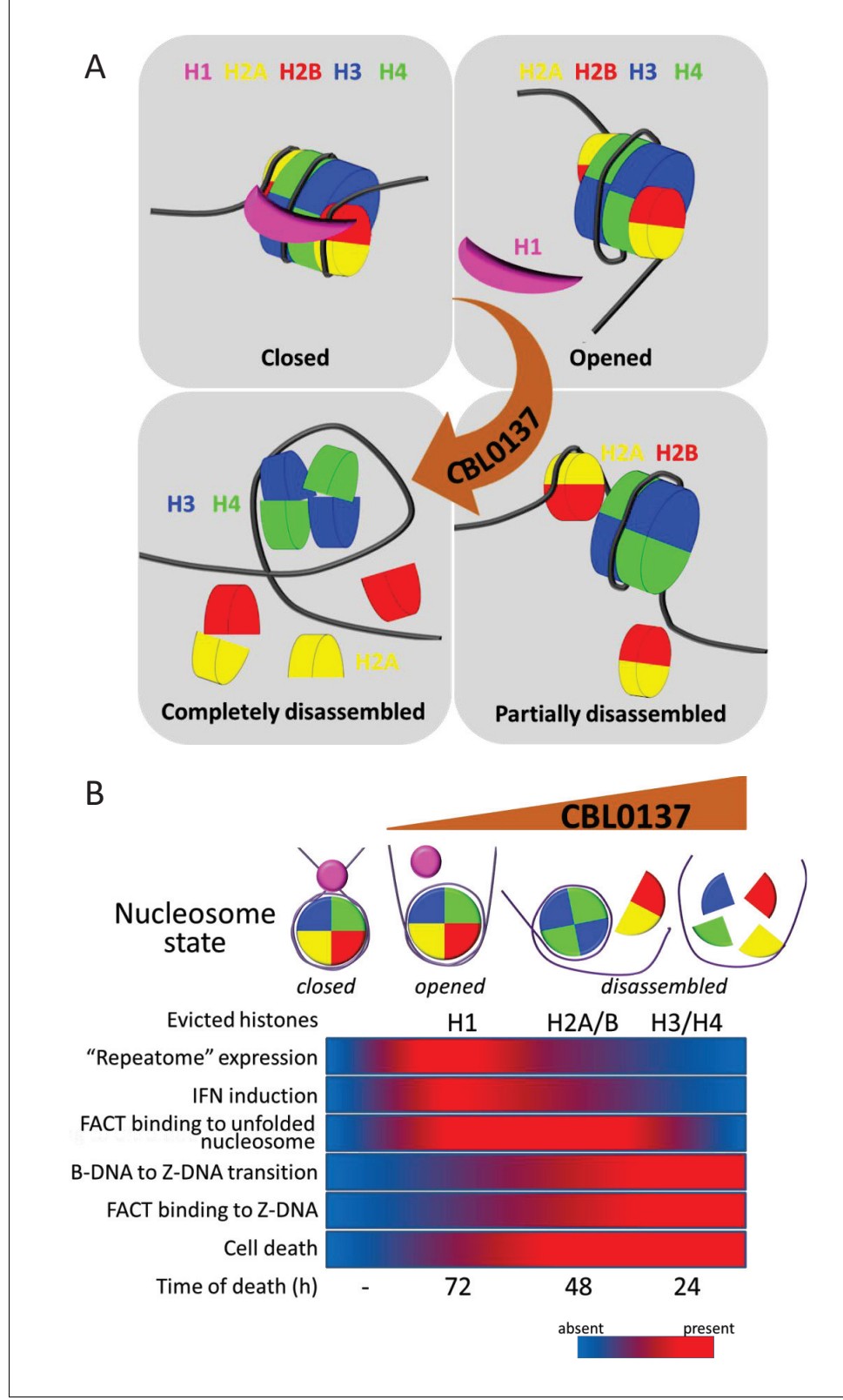

**Figure 10.** Proposed scheme of nucleosome states in the absence and presence of increasing concentrations of CBL0137 (**A**) and associated biological processes in cells (**B**).
DOI: https://doi.org/10.7554/eLife.30842.025

CBL0137, the drug can be washed from the cells with almost no harm to the cells; cells incubated with 1–3 µM CBL0137 for no more than 1 hr survive with some loss of viability. However, treatment with >5 µM CBL0137 for more than 15 min leads to complete cell death after 48 hr, which is in line with the complete disassembly of chromatin (*Figure 10B*). An important and completely unknown question is what causes cell death upon loss of chromatin integrity.

Other cellular 'responses' to CBL0137 treatment are also aligned with a gradual change in nucleosome organization. The first cellular factor that we found, 'reacting' to the problems with chromatin packaging in CBL0137-treated cells, was histone chaperone FACT, which underwent c-trapping in response to nucleosome disassembly (*Safina et al., 2017*). It was shown that human FACT cannot bind and uncoil a 'closed' nucleosome (*Tsunaka et al., 2016*; *Valieva et al., 2017*) (*Figure 10*). However, nucleosome opening most probably allows FACT to invade the nucleosome and bind the H3/H4 tetramer via the middle domain of SPT16 and the H2A/H2B dimer via the c-terminal portions of both subunits (*Belotserkovskaya et al., 2003*; *Kemble et al., 2015*; *Tsunaka et al., 2016*; *Winkler and Luger, 2011*). In cells, we see this as SPT16-mediated 'n (nucleosome)-trapping,' which is observed at 0.3–1 µM CBL0137. At CBL0137 concentrations greater than 3 µM, significant negative supercoiling, as a result of nucleosome disassembly, caused base-unpairing of DNA and the transition of B-DNA into left-handed Z-form DNA, which is bound by FACT via the c-terminal intrinsically disordered domain of SSRP1 subunit (*Safina et al., 2017*). FACT has a plethora of activities related to transcription (rev. in [*Reinberg and Sims, 2006*], [Gurova and Studitsky, 2018; in preparation). Therefore, many CBL0137 effects on transcription might be explained by the functional inactivation of FACT. However, we suspected that opening of chromatin might have consequences independent on FACT.

The most prominent and uniform change observed in the presence and absence of FACT was activation of the IFN response in all tested organs and cells at both the mRNA and protein levels. The induction of IFN-sensitive genes (ISG) has a bell-shaped curve, with a maximum increase at a CBL0137 concentration of approximately 0.5–1 µM, which corresponds to nucleosome 'opening' followed by the inhibition of transcription at concentrations greater than 1 µM, when nucleosome disassembly starts. We believe that IFN is activated in response to the dsRNA formed due to the increased transcription from normally silenced constitutive heterochromatin. There was a report that proposed the role of histone H1 in the silencing of ISGs where knockdown of the H1 chaperone TAF1 in cells resulted in H1 loss from ISG promoters and activation of their transcription (*Kadota and Nagata, 2014*). However, the set of data in that report does not exclude the role of general chromatin decondensation due to the reduction of H1 deposition in heterochromatin. We showed that in the case of CBL0137, transcription of ISGs is abrogated if the IFN signaling pathway is disabled (e.g. *Ifnar1-/-* cells and mice), suggesting that there is no direct effect of CBL0137 on the promoters of ISGs. If we assume that IFN is activated in response to increased transcription of 'repeatome', then the puzzling and intriguing question is why nucleosome 'opening' is accompanied by selective activation of transcription of these elements, but not most genes, while nucleosome disassembly leads to the inhibition of transcription genome-wide. We propose that displacement of histone H1 may make heterochromatin accessible for RNA polymerase, while accessibility of genomic regions not locked with H1 does not change. The inhibition may be explained by a direct effect of higher concentrations of CBL0137, bound to DNA, on RNA polymerase activity, but this requires further investigation.

Importantly, the anticancer effect in mice is seen at concentrations that activate the IFN response, suggesting that chromatin opening without chromatin disassembly is enough to cause an anticancer effect in vivo. Activation of the IFN response by CBL0137 also creates an opportunity to use IFN-responsive factors as pharmacodynamic markers of curaxin activity in the clinic, since many of these factors are secreted and were detected in the plasma of mice.

Thus, curaxins, compounds with established anticancer activity in multiple preclinical models, may also be used as a tool to affect chromatin organization and folding globally in cells and animals to study the consequences of an abrupt loss of epigenetic regulation in vivo. Potentially, this property of curaxins may be useful for reprogramming and other types of therapeutic epigenetic manipulations.

One more important aspect of this study was the expansion of the set of situations when the TRAIN response occurs. Previously, we and others observed TRAIN upon loss of DNA methylation (*Leonova et al., 2013*), (*Chiappinelli et al., 2015*; *Guryanova et al., 2016*; *Roulois et al., 2015*),

mostly in the case of p53 deficiency (*Leonova et al., 2013*). Here, we observed that general chromatin decondensation due to nucleosome opening is also accompanied by TRAIN. Based on this data, we propose that TRAIN may be a more general response to the problems associated with chromatin packaging and the danger of the loss of epigenetic integrity. The stability of the cell differentiation status, which can still be easily detected even in HeLa cells after more than 60 years in culture, is a long-standing puzzle. Theoretically, stability can be ensured either by the low rate of error in the replication of chromatin marks or by an auto-regulatory mechanism that eliminates cells with lost epigenetic information, similar to p53-dependent apoptosis in response to DNA damage. It is tempting to speculate that TRAIN may be such a mechanism, triggered by elevated expression of silenced repetitive genomic elements and executed by the IFN response, which is capable of killing cells. It is likely that this mechanism evolved in response to real viral threats, especially in the form of the expansion of endogenous retroviral elements. This mechanism may serve as an epigenetic lock, which supports the differentiation state and limits cell plasticity and, therefore, the ability to transdifferentiate and regenerate organs and tissues, as well as to undergo epigenetic transitions associated with malignant transformation and tumor progression.

## Materials and methods

### Chemicals, reagents, and plasmids

CBL0137 was provided by Incuron, LLC. Trichostatin A was purchased from Sigma-Aldrich. Poly I:C (polyinosinic:polycytidylic acid) was purchased from Tocris Bioscience. ISRE-mCherry and ISRE-Luc reporter plasmids was purchased from Cellecta. Two *ZBP1* gRNA, control gRNA, and CRISPR/Cas9 lentiviral single vector plasmids and mCherry-tagged histone H1.5 were purchased from GeneCopoeia. GFP-tagged SSRP1 was previously described in *Safina et al. (2017)*.

### Cells

HeLa, HepG2, HT1080and normal skin fibroblast (NDF) cells were obtained from ATCC and maintained in DMEM with 5% Fetal Bovine Serum (FBS) and 50 µg/ml of penicillin/streptomycin. Mouse squamous cell carcinoma SCCVII cells were obtained from Dr. Burdelya (Roswell Park Cancer Institute (RPCI), Buffalo, NY). Preparation and maintenance of HeLa-TI cell populations containing integrated avian sarcoma genome with silent green fluorescent protein (GFP) gene were described previously (*Poleshko et al., 2010*; *Shalginskikh et al., 2013*).

### Isolation and cultivation of mouse embryonic fibroblast (MEF) cells

MEFs were isolated from embryos at 13.5 days post-coitus by breeding C57BL/6 and *Trp53*-heterozygous (*Trp53$^{+/-}$*) mice or C57BL/6 Ifnar-null and *Trp53*-heterozygous females and males as described in the protocol http://www.molgen.mpg.de/~rodent/MEF_protocol.pdf. Cells from each embryo were individually plated. DNA was isolated from individual embryos using the PureLink$^{TM}$ Genomic DNA Kit (Invitrogen, Carlsbad, CA). The genotype of the embryos was determined by PCR with the following primers: *Trp53$^{+/+}$* ACAGCGTGGTGGTACCTTAT; *Trp53$^{+/-}$* TATACTCAGAGCCGGCCT; *Trp53$^{-/-}$* TCCTCGTGCTTTACGGTATC; *Ifnar1-/-* 5'common 5'-ATTATTAAAAGAAAAGACGAGGAGGCGAAGTGG-3'; *Ifnar1-/-* 3'WT allele 5'-AAGATGTGCTGTTCCCTTCCTCTGCTCTGA-3'; *Ifnar1-/-* Neo 5'-CCTGCGTGCAATCCATCTTG-3'.

MEF cells were cultured in DMEM supplemented with 10% FBS and 50 µg/ml penicillin/streptomycin for no more than eight passages. Transformation of p53-/- cells was done via lentiviral delivery of HRasV12 oncogene followed by selection with bleomycin.

### Reverse transcriptase-PCR

cDNA was made from 1 µg total RNA isolated by TRIzol (Invitrogen) using the iScript cDNA Synthesis Kit (BioRad, Hercules, CA). The PCR reaction was carried out in accordance with the manufacturer's protocol using USB Taq PCR Master Mix (Affymetrix, Santa Clara, CA) in a reaction volume of 25 µl with 100 ng of the following primers: mouse *Irf7* sense: 5'-CAGCCAGCTCTCACCGAGCG; antisense: 5'-GCCGAGACTGCTGCTGTCCA, mouse *Ifit*3 (Isg49) sense: 5'-GCCGTTACAGGGAAATACTGG; antisense: 5'-CCTCAACATCGGGGCTCT; human *IFIT3* (ISG56) sense: 5'-CCCTGCAGAACGGCTGCCTA; antisense: 5'-AGCAGGCCTTGGCCCGTTCA; mouse *Isg15* sense: 5'-

AAGAAGCAGATTGCCCAGAA; antisense: 5'- TCTGCGTCAGAAAGACCTCA; mouse *Usp18* sense: 5'-AAGGACCAGATCACGGACAC; antisense: 5'- CACATGTCGGAGCTTGCTAA, mouse Actb sense: 5'- GCTCCGGCATGTGCAA; antisense: 5'-AGGATCTTCATGAGGTAGT-3', human *ZBP1* sense: 5'- TGCAGCTACAATTCCAGAGA; antisense: 5'- GAAGGTGCCTGCTCTTCATC.

## Western immunoblotting

Protein extracts were prepared by lysing cells or tissues in RIPA buffer (Sigma-Aldrich, St Louis, MO) containing protease inhibitor cocktail (Sigma-Aldrich). Extracts were spun down at 10,000 x rpm for 10 min at 4°C to obtain the soluble fraction. Protein concentrations were determined using the Bio-Rad Protein Assay (Bio-Rad). Equal amounts of protein were run on 4–20% precast gradient gels (Invitrogen) and blotted/transferred to Immobilon-P membrane (Millipore, Burlington, MA). Membranes were blocked with 5% non-fat milk-TBS-T buffer for 1 hr and incubated overnight with primary antibodies. The following antibodies were used: 1:10,000 anti-ISG49 and anti-ISG56 (a kind gift from Dr. Ganes Sen, Cleveland Clinic, Cleveland, OH). β-Actin (Santa Cruz Biotechnology, Santa-Cruz, CA) antibodies were used to verify equal protein loading and transfer. Anti-mouse and anti-rabbit secondary horseradish peroxidase-conjugated antibodies were purchased from Santa Cruz. ECL detection reagent (GE Healthcare, Little Chalfont, UK) was used for protein visualization on autoradiography film (Denville Scientific, Holliston, MA).

## Northern hybridization

Mouse cDNA probes for SINE B1 (sense: 5'-GCCTTTAATCCCAGCACTTG, antisense: 5'-CTCTGTG TAGCCCTGGTCGT), were made by reverse-PCR from the total RNA of MEF cells. The cDNAs were labeled with [$\alpha^{32}$P]-dCTP, using the Random Primed DNA Labeling Kit following the manufacturer's protocol (Roche, Basel, Switzerland). Total RNA was extracted from cells that were either untreated or treated with 10 μM of 5-aza-dC for 48 hr or 0.5 uM of CBL0137 for 24 hr using Trizol (Invitrogen). Total RNA (5 μg) was loaded onto each lane, electrophoresed in an agarose–formaldehyde gel, and transferred onto a Hybond-N membrane (Amersham Pharmacia Biotech, Little Chalfont, UK). After UV crosslinking, the transfers were hybridized with [$\alpha^{32}$P]-dCTP-labeled probes and analyzed by autoradiography at −80°C.

## Immunofluorescent staining

Cells were plated in 35-mm glass bottom plates from MatTek Corporation (Ashland, MA). After treatment, cells were washed with PBS and fixed in 4% paraformaldehyde at room temperature for 15 min. For Z-DNA staining, a 4% paraformaldehyde solution containing 0.1% Triton-X100 in PBS was added to cells for 15 min immediately after removal of media. Cells were then washed three times with PBS. Blocking was done in 3% BSA, 0.1% Triton-X100 in PBS. Primary antibodies: Z-DNA from Abcam (Cambridge, UK, cat# ab2079) was used at 1:200 dilution. dsRNA antibody (J2) from Scicons (Hungary) was used at 1:50 dilution. AlexaFluor 488 or 594 donkey anti-mouse (Invitrogen, cat# A21206; 1:1000) and AlexaFluor 594 donkey anti-sheep (Jackson ImmunoResearch, West Groove, PA, cat# 713-585-147; 1:500) were used as secondary antibodies. Antibodies were diluted in 0.5% BSA +0.05% Triton X100 in PBS. After each antibody incubation, cells were washed three times with 0.05% Triton X100 in PBS. For DNA counterstaining, 1 μg/ml solution of Hoechst 33342 in PBS was used. Images were obtained with a Zeiss Axio Observer A1 inverted microscope with N-Achroplan 100×/1.25 oil lens, Zeiss MRC5 camera, and AxioVision Rel.4.8 software.

## Flow cytometry

Flow cytometry was performed on LSR Fortessa A and BD LSRII UV A Cytometers (BD Biosciences, San Jose, CA). Obtained data were analyzed by the WinList 3D program (Verity Software House, Topsham, ME).

## Microarray hybridization, RNA-sequencing, and analyses

Total RNA from cells was isolated using TRIzol reagent (Invitrogen). Two biological replicates of each condition were used for microarray hybridization and RNA-sequencing. RNA processing, labeling, hybridization, generation of libraries, and sequencing were performed by the Genomics Shared Resource (Roswell Park). MouseWG-6 v2.0 Expression BeadChip array (Illumina, San Diego, CA) was

used for hybridization. The TruSeq Stranded Total RNA Library Prep Kit with Ribo-Zero Mouse kit (Illumina) was used for library preparation. Sequencing was done using Illumina HiSeq 2000 system. Gene expression data were analyzed using GeneSpring GX for microarray hybridization and Strand NGS for RNA-sequencing (Agilent Genomics, Santa Clara, CA). GSEA was done using MSigDB (Broad Institute, Cambridge, MA). Defaults statistical methods recommended by the programs were used.

The GRCm38.p5 (Genome Reference Consortium Mouse Reference 38) assembly was used for all mouse data, and the set of annotated repeat types was taken from the Repbase database (*Kapitonov and Jurka, 2008*). Sailfish version 0.6.3 aligner was used to quantify the abundance of previously annotated RNA transcripts from RNA-seq data (*Patro et al., 2014*). Analyses were performed with a modified k-mer size of k = 17 and used 16 concurrent threads (p 16). Sailfish was run with the poly-A option, which discards k-mers consisting of k consecutive A's or T's, and bias correction was enabled. Importantly, a read was associated with a particular 'repeat' type if it satisfied the following criteria: (a) The read aligned to a single or multiple locations within the canonical sequence of that repeat type, or annotated instances of that repeat type within the genome, incorporating 13 bp (half of the read length) genomic sequences flanking the annotated instances. (b) No alignment of such or better quality to canonical or instance sequences associated with any other annotated repeat types could be obtained. An optional masking procedure, designed to exclude reads potentially originating from un-annotated repeat types, added a third requirement. (c) No alignment of such or better quality to any portion of the genome assembly that is not associated with the annotated instances of that repeat type could be obtained. Collectively, the combined repeat assembly file contained a single FASTA entry for each repeat type as defined in the Repbase database. The Transcripts per Million quantification number was computed as described in (*Wagner et al., 2013*), (*Qian and Huang, 2005*) and is meant as an estimate of the number of transcripts, per million observed transcripts, originating from each target transcript. Its benefit over the F/RPKM measure is that it is independent of the mean expressed transcript length (i.e. if the mean expressed transcript length varies between samples, for example, this alone can affect differential analysis based on the K/RPKM). Finally, we used the variational Bayesian EM algorithm integrated into sailfish, rather than the 'standard' EM algorithm, to optimize abundance estimates. p-Values were corrected using an FDR correction using Baggerley test {*Baggerly et al. (2003)* #450} for samples with replicates (wild-type MEF and lung) and Kal's Z-test {*Kal et al. (1999)* #449} on samples without replicates (p53-/-MEF).

## Statistical analyses

Unpaired t-test was used for comparison of quantitative data between control and experimental groups. Analyses were conducted using SAS v9.4 (Cary, NC) and all p-values are two-sided.

## Animal experiments

All experiments were reviewed and approved by the Roswell Park IACUC. Mice were maintained in the Laboratory Animal Shared Resource with controlled air, light, and temperature conditions, and fed ad libitum with free access to water. NIH Swiss and C57Bl/6J mice were purchased from The Jackson Laboratory (Bar Harbor, ME).

For the analysis of gene expression, groups of two mice (male or female, 6–8 weeks old) were treated intravenously with one dose of CBL0137 dissolved in 5% dextrose using the tail vein. Control mice received an injection of 5% dextrose.

For the comparison of the efficacy of different doses of CBL0137, SCID mice (male, 6–8 weeks old) were inoculated in both flanks with $10^6$ HepG2 cells in PBS. When tumors reach ~50 mm$^3$, mice were randomized into four groups (n = 5). Sample size was calculated by power analysis (*Festing and Altman, 2002*) using power of 95%, expected difference in control and treated groups of two-folds and standard deviation of 40% of mean value. Treatment was administered intravenously once a week.

For the comparison of CBL0137 anti-cancer efficacy depending on the presence of IFN response, groups of C57Bl/6 mice with wild type *Ifnar1* or *Ifnar1-/-* were inoculated subcutaneously into one flank with 750,000 p53-/- MEF cells, *Ifnar1* wild type or *Ifnar1-/-* transformed with HRasV12, 10 mice for each combination of mouse and tumor genotypes. Tumors were measured every day. When

tumor in any mouse reached 50 mm$^3$, this mouse was assigned to treatment group, first for 50 mg/kg CBL0137, second for vehicle (5% dextrose) and so on until five mice were accumulated in each group. Treatment was done via tail vein injection once a week for 3 weeks.

## Acknowledgements

This work was supported by Incuron LLC (KG), and by National Cancer Institute grants R01CA197967 (KG) and P30CA016056 (Roswell Park Cancer Center). Analysis of CBL0137 epigenetic activity in HeLa-Ti cells was supported by Russian Science Foundation (17-15-01526).

## Additional information

### Funding

| Funder | Grant reference number | Author |
| --- | --- | --- |
| National Cancer Institute | RO1CA197967 | Katerina Gurova |
| National Cancer Institute | R21CA198395 | Katerina Gurova |
| Russian Science Foundation | 17-15-01526 | Marianna G Yakubovskaya |
| Roswell Park Cancer Institute | P30CA016056 | Katerina Gurova |
| Incuron LLC | | Katerina Gurova |

The funders had no role in study design, data collection and interpretation, or the decision to submit the work for publication.

### Author contributions

Katerina Leonova, Data curation, Formal analysis, Investigation, Methodology, Writing—original draft, Project administration; Alfiya Safina, Data curation, Formal analysis, Supervision, Visualization, Methodology; Elimelech Nesher, Data curation, Investigation, Writing—review and editing; Poorva Sandlesh, Rachel Pratt, Brittany Lipchick, Ilya Gitlin, Investigation; Catherine Burkhart, Investigation, Methodology, Writing—review and editing; Costakis Frangou, Data curation, Software, Formal analysis; Igor Koman, Conceptualization, Supervision, Funding acquisition, Writing—review and editing; Jianmin Wang, Data curation, Formal analysis, Visualization, Writing—review and editing; Kirill Kirsanov, Conceptualization, Validation, Investigation, Methodology, Writing—review and editing; Marianna G Yakubovskaya, Conceptualization, Data curation, Formal analysis, Supervision, Funding acquisition, Validation, Writing—review and editing; Andrei V Gudkov, Conceptualization, Data curation, Formal analysis, Supervision, Writing—review and editing; Katerina Gurova, Conceptualization, Data curation, Formal analysis, Supervision, Funding acquisition, Validation, Investigation, Visualization, Methodology, Writing—original draft, Project administration, Writing—review and editing

### Author ORCIDs

Elimelech Nesher http://orcid.org/0000-0002-8326-5535
Katerina Gurova http://orcid.org/0000-0001-9189-0712

### Ethics

Animal experimentation: This study was performed in strict accordance with the recommendations in the Guide for the Care and Use of Laboratory Animals of the National Institutes of Health. All of the animals were handled according to approved institutional animal care and use committee (IACUC) protocols (#1093M) of Roswell Park Cancer Institute. The protocol was approved by the Committee on the Ethics of Animal Experiments of Roswell Park Cancer Institute.

### Decision letter and Author response

Decision letter https://doi.org/10.7554/eLife.30842.030
Author response https://doi.org/10.7554/eLife.30842.031

## Additional files

### Supplementary files
• Transparent reporting form
DOI: https://doi.org/10.7554/eLife.30842.026

### Major datasets
The following dataset was generated:

| Author(s) | Year | Dataset title | Dataset URL | Database, license, and accessibility information |
|---|---|---|---|---|
| Gurova K | 2017 | Effect of CBL0137 on gene expression in mouse cells and tissues | https://www.ncbi.nlm.nih.gov/geo/query/acc.cgi?acc=GSE102768 | Publicly available at the NCBI Gene Expression Omnibus (accession no: GSE102768) |

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
