## [Decision Letter]

Thank you for submitting your article "TRAIN in response to chromatin destabilization induced with anti-cancer small molecules" for consideration by *eLife*. Your article has been reviewed by two peer reviewers, and the evaluation has been overseen by a Reviewing Editor and Kevin Struhl as the Senior Editor. The following individual involved in review of your submission has agreed to reveal his identity: Robert H Silverman (Reviewer #1).

The reviewers have discussed the reviews with one another and the Reviewing Editor has drafted this decision to help you prepare a revised submission.

In this manuscript, Leonova et al. report that the anticancer molecule curaxin destabilizes nucleosomes resulting in TRAIN. This group and others previously showed that demethylation of DNA results in transcription of repetitive elements that generate dsRNA and an IFN response. The authors now demonstrate that the tumor suppressive activity of curaxin is partially independent of p53 through TRAIN. They also indicate that it is independent of FACT, which had previously been identified as a putative target. Using microarray analysis, Leonova et al. find that curaxin induces the IFN pathway in the liver, lung, and spleen, consistently. They go on to show that several ISGs are upregulated on the order of hours rather than days in mouse tissues as well as mouse and human cell lines. The authors also demonstrate that this induction does not happen in transformed cell lines. The authors next show that increased IFN signaling in response to CBL0137 is associated with increased levels of dsRNA, potentially from transcription of repetitive elements, some of which are induced to similar levels as with HDAC inhibitors. Finally, the authors show that in a Ras-driven model, colony formation is impeded in the absence of p53, but not in the absence of p53 and IFNAR1.

After discussion, the reviewers assessed that, in its present form, the manuscript did not provide enough new knowledge relative to the previous manuscripts by these teams on CBL0137 and TRAIN, as IFN induction by derepression of heterochromatin, and subsequent production of dsRNAs, has been demonstrated by this group and others in a number of settings. In view of the severity of these concerns, we would welcome a quick response from you as to the feasibility of the essential additional work. Please document your ability to perform the additional animal work within a reasonable length of time in order to persuade the reviewers that recommended "revision" is justified.

If these experiments will require a substantial effort, it may be best to withdraw your submission to a future date or to another journal.

Major Concerns:

1) The main concern is that there is little if any proof that TRAIN is actually required for the anti-tumor properties of curaxin. The last figure of the paper hints at this by testing the impact of depletion of the IFNA receptor (in a p53 null background) on RAS-transformed MEF foci. The authors conclude that TRAIN 'impacts on the tumor-preventive activity of curaxin', but there MEF foci formation assays cannot be equated to tumorigenesis. Given that the authors have already involved FACT and p53 as potential mediators of drug action, what is the relative importance of TRAIN in the potential therapeutic action of the drug? The reviewers considered this to be a critical question that would require new experiments in animal models to assess the relative contribution of each of the pathways to the therapeutic action of the drug.

2) Another major concern is regarding the conclusion that 'TRAIN occurs in response to chromatin opening'. This conclusion if unfounded, as a cause-effect relationship between curaxin, HDAC inhibitors, chromatin opening and TRAIN cannot be established. Curaxin and HDAC inhibitors could be acting by other mechanisms beyond loosening nucleosomes to activate TRAIN. To prove that nucleosome opening is the key intermediate in the 'signaling pathway' seems impossible to demonstrate experimentally. As such, there is a correlation here, but no cause-effect relationship. The manuscript should be carefully rewritten to avoid making cause-effect relationships from these correlative observations.

3) Figure 1 and Figure 2 show that CBL0137 up-regulates ISGs in vivo in a FACT- independent manner; by examining 2 FACT-positive and 2 FACT-negative mouse tissues. Three doses of CBL0137 were used correlating to "weak, intermediate, and strong" anti-cancer activity. However, the lowest dose used, 30mg/kg actually increased tumor growth (as shown in Figure 1—figure supplement 1). Some explanation for this should be stated in the text. The next two highest doses of CBL0137 (60 and 90mg/kg) did reduce tumor growth.

4) Only one gene (IFIT3) was up-regulated in all four organs (lung, spleen, liver, testis) by the Venn diagram, very few genes were affected in testis (Figure 1), yet IFITm3 was also up-regulated in all four organs (Figure 1). Why is IFITm3 not in the Venn diagram center?

5) There was no mention in the text that IFITm3 is unrelated to IFIT3, although levels of IFITm3 are reported in Figure 1. Some explanation for including IFITm3 in Figure 1 should be provided in the text, while mentioning IFIT3 and IFITm3 are unrelated IFN inducible proteins.

6) CBL0137 induced an IFN response in normal mouse and human cells, but not in tumor cells. However, using an ISRE-reporter construct, activation of the ISRE could be detected indirectly in both mouse and human tumor cells (Figure 4). It is suggested that failure to detect ISG up-regulation in tumor cells is due to inactivation of the IFN response. However, activation of ISREs should lead to up-regulation of endogenous ISGs, not just of the ISRE-reporter construct. Further explanation should be provided to account for this discrepancy. Is the reporter construct more sensitive to IFN than endogenous ISGs? A comparison of different low doses of exogenous IFN on the ISRE-reporter vs. endogenous ISGs might resolve this issue. This should be done on normal and tumor cells.

7) CBL0137 treatment greatly up-regulated levels of dsRNA from repetitive elements in the genome, including SINE B1 and LINE1 (Figure 6). CBL0137 treated MEF p53-/- Ifnar-/- and HT1080 are convincing, but not the MEF p53-/-, perhaps the authors have better images of this (Figure 6, left two panels)?

8) There does not appear to be much of an up-regulation of the repetitive transcripts in the lungs of p53+/+ mice in Figure 6, this should be mentioned in the text with an explanation or statistics showing significance provided (add p-values for all of Figure 6). The location of Table S7, referred to in the text, is not clear labeled and could be not be evaluated.

---

## [Author Response]

Major Concerns:1) The main concern is that there is little if any proof that TRAIN is actually required for the anti-tumor properties of curaxin. The last figure of the paper hints at this by testing the impact of depletion of the IFNA receptor (in a p53 null background) on RAS-transformed MEF foci. The authors conclude that TRAIN 'impacts on the tumor-preventive activity of curaxin', but there MEF foci formation assays cannot be equated to tumorigenesis. Given that the authors have already involved FACT and p53 as potential mediators of drug action, what is the relative importance of TRAIN in the potential therapeutic action of the drug? The reviewers considered this to be a critical question that would require new experiments in animal models to assess the relative contribution of each of the pathways to the therapeutic action of the drug.

We performed the following experiment to address this concern. We transformed MEF from *Ifnar1-/-* and wild-type C57BL/6 mice with mutant Ras (H-RasV12) and the p53 inhibitor, GSE56. The transformed cells were inoculated into either *Ifnar1-/-* or wild-type animals. Our goal was to treat all four groups with either vehicle control or CBL0137 to determine whether the efficacy of CBL0137 was dependent on the presence of Ifnar either in the tumors or the mice. This experiment demonstrated a significant difference in the growth of the tumors depending on their *Ifnar1* status. The fastest and most consistent tumor growth was observed for *Ifnar1-/-* tumors growing in *Ifnar1-/-* mice. The wild-type tumors grew approximately four times slower than the *Ifnar1-/-* tumors in *Ifnar1-/-* mice. Although the *Ifnar1-/-* tumors grew similarly in wild-type mice, only two tumors out of the five grew. Finally, wild-type tumors did not grow beyond 50-100 mm^3^ during two months of observation. These data suggest that the IFN response plays an important role as a tumor suppressor. It is likely that this role is independent of its role in the immune system because the difference was observed on an *Ifnar1-/-* background. Based on the growth of these different tumor lines in mice, we were only able to compare the efficacy of CBL0137 in *Ifnar1-/-* mice. CBL0137 inhibited the growth of the wild-type tumors but only slowed down the growth of the *Ifnar1-/-* tumors, demonstrating that the IFN response plays a role in the anti-cancer activity of CBL0137. We included these data as Figure 9.

2) Another major concern is regarding the conclusion that 'TRAIN occurs in response to chromatin opening'. This conclusion if unfounded, as a cause-effect relationship between curaxin, HDAC inhibitors, chromatin opening and TRAIN cannot be established. Curaxin and HDAC inhibitors could be acting by other mechanisms beyond loosening nucleosomes to activate TRAIN. To prove that nucleosome opening is the key intermediate in the 'signaling pathway' seems impossible to demonstrate experimentally. As such, there is a correlation here, but no cause-effect relationship. The manuscript should be carefully rewritten to avoid making cause-effect relationships from these correlative observations.

We have modified the manuscript as suggested.

3) Figure 1 and Figure 2 show that CBL0137 up-regulates ISGs in vivo in a FACT- independent manner; by examining 2 FACT-positive and 2 FACT-negative mouse tissues. Three doses of CBL0137 were used correlating to "weak, intermediate, and strong" anti-cancer activity. However, the lowest dose used, 30mg/kg actually increased tumor growth (as shown in Figure 1—figure supplement 1). Some explanation for this should be stated in the text. The next two highest doses of CBL0137 (60 and 90mg/kg) did reduce tumor growth.

The reviewer is correct. We have changed the text to “absent, moderate, and strong.” The incorrect interpretation was based on previous data from different tumor models where a dose of 30 mg/kg administered daily demonstrated anti-cancer activity. The study presented in the paper was performed with a once per week CBL0137 treatment regimen.

4) Only one gene (IFIT3) was up-regulated in all four organs (lung, spleen, liver, testis) by the Venn diagram, very few genes were affected in testis (Figure 1), yet IFITm3 was also up-regulated in all four organs (Figure 1). Why is IFITm3 not in the Venn diagram center?

We used the criteria of fold change 1.5 and p-value <0.05 to build the lists of genes for which expression changed in any of the conditions. *Ifitm3* did not pass the fold change criterion in liver (i.e., maximum change in liver was 1.45-fold). We added these details to the text and the corresponding figure legend.

5) There was no mention in the text that IFITm3 is unrelated to IFIT3, although levels of IFITm3 are reported in Figure 1. Some explanation for including IFITm3 in Figure 1 should be provided in the text, while mentioning IFIT3 and IFITm3 are unrelated IFN inducible proteins.

We have added these explanations to the revised manuscript.

6) CBL0137 induced an IFN response in normal mouse and human cells, but not in tumor cells. However, using an ISRE-reporter construct, activation of the ISRE could be detected indirectly in both mouse and human tumor cells (Figure 4). It is suggested that failure to detect ISG up-regulation in tumor cells is due to inactivation of the IFN response. However, activation of ISREs should lead to up-regulation of endogenous ISGs, not just of the ISRE-reporter construct. Further explanation should be provided to account for this discrepancy. Is the reporter construct more sensitive to IFN than endogenous ISGs? A comparison of different low doses of exogenous IFN on the ISRE-reporter vs. endogenous ISGs might resolve this issue. This should be done on normal and tumor cells.

To investigate this discrepancy, we measured ISRE reporter activity in parallel with RT-PCR for ISGs in human tumor cell lines and MEF cells treated with a range of IFNα and CBL0137 doses. We found that lower IFNa doses, which were already able to activate the reporter in the tumor cells, were not always able to induce the expression of IFIT3 or IRF7. Expression of IFIT3 or IRF7 only occurred at higher IFN doses. These data suggest that the reporter assay is more sensitive than the RT-PCR assay used, what may be explained by the presence of five ISRE elements in the reporter, while endogenous genes have just one. We added some of this information as Figure 4—figure supplement 1 (new).

7) CBL0137 treatment greatly up-regulated levels of dsRNA from repetitive elements in the genome, including SINE B1 and LINE1 (Figure 6). CBL0137 treated MEF p53-/- Ifnar-/- and HT1080 are convincing, but not the MEF p53-/-, perhaps the authors have better images of this (Figure 6, left two panels)?

We have included better images as recommended.

8) There does not appear to be much of an up-regulation of the repetitive transcripts in the lungs of p53+/+ mice in Figure 6, this should be mentioned in the text with an explanation or statistics showing significance provided (add p-values for all of Figure 6). The location of Table S7, referred to in the text, is not clear labeled and could be not be evaluated.

We have provided the requested information and changed the text accordingly.